# Polymorphism in the aggressive mimicry lure of the parasitic freshwater mussel *Lampsilis fasciola*

Trevor L. Hewitt[1], Paul D. Johnson[2], Michael Buntin[2], Talia Y. Moore[1,3] and Diarmaid Ó Foighil[1]

[1] Ecology and Evolutionary Biology, University of Michigan—Ann Arbor, Ann Arbor, Michigan, United States
[2] Alabama Aquatic Biodiversity Center, Marion, Alabama, United States of America
[3] Robotics Department & Mechanical Engineering Department, University of Michigan—Ann Arbor, Ann Arbor, Michigan, United States

Corresponding author
Trevor L. Hewitt,
htrevor@umich.edu

## ABSTRACT

Unionoid freshwater mussels (Bivalvia: Unionidae) are free-living apart from a brief, obligately parasitic, larval stage that infects fish hosts, and gravid female mussels have evolved a spectrum of strategies to infect fish hosts with their larvae. In many North American species, this involves displaying a mantle lure: a pigmented fleshy extension that acts as an aggressive mimic of a host fish prey, thereby eliciting a feeding response that results in host infection. The mantle lure of *Lampsilis fasciola* is of particular interest because it is apparently polymorphic, with two distinct primary lure phenotypes. One, described as "darter-like", has "eyespots", a mottled body coloration, prominent marginal extensions, and a distinct "tail". The other, described as "worm-like", lacks those features and has an orange and black coloration. We investigated this phenomenon using genomics, captive rearing, biogeographic, and behavioral analyses. Within-brood lure variation and within-population phylogenomic (ddRAD-seq) analyses of individuals bearing different lures confirmed that this phenomenon is a true polymorphism. The relative abundance of the two morphs appears stable over ecological timeframes: the ratio of the two lure phenotypes in a River Raisin (MI) population in 2017 was consistent with that of museum samples collected at the same site six decades earlier. Within the River Raisin, four main "darter-like" lure motifs visually approximated four co-occurring darter species (*Etheostoma blennioides*, *E. exile*, *E. microperca*, and *Percina maculata*), and the "worm-like" lure resembled a widespread common leech, *Macrobdella decora*. Darters and leeches are typical prey of *Micropterus dolomieui* (smallmouth bass), the primary fish host of *L. fasciola*. *In situ* field recordings of the *L. fasciola* "darter" and "leech" lure display behaviors, and the lure display of co-occurring congener *L. cardium*, were captured. Despite having putative models in distinct phyla, both *L. fasciola* lure morphs have largely similar display behaviors that differ significantly from that of sympatric *L. cardium* individuals. Some minor differences in the behavior between the two *L. fasciola* morphs were observed, but we found no clear evidence for a behavioral component of the polymorphism given the criteria measured. Discovery of discrete within-brood inheritance of the lure polymorphism implies potential control by a single genetic locus and identifies

*L. fasciola* as a promising study system to identify regulatory genes controlling a key adaptive trait of freshwater mussels.

## INTRODUCTION

In ecology, mimicry refers to a convergent adaptive trait prevalent in many biological communities: the deceptive resemblance of one organism to another (*Pasteur, 1982*; *Schaefer & Ruxton, 2009*; *Maran, 2015*). It involves three categories of interacting ecological players: mimic (organism displaying the deceptive resemblance), model (organism being mimicked), and receiver (organism being deceived) (*Pasteur, 1982*; *Maran, 2015*). Mimicry occurs across a wide variety of ecological contexts and sensory modalities, but conceptually (*Jamie, 2017*), individual cases can be categorized by the traits being mimicked (signals or cues), as well as by the degree of deceptiveness (aggressive, rewarding, Müllerian or Batesian mimicry). Mimicry is also ubiquitous throughout nature, with many prominent well studied examples including mantids (*O'Hanlon, Holwell & Herberstein, 2014*), spiders (*Ceccarelli, 2013*), fish (*Randall, 2005*), and many more.

Mimetic systems that are polymorphic (multiple within-species mimic morphs with discrete models) have been particularly influential in uncovering the genetic basis of complex adaptive traits in natural populations (*Clarke, Sheppard & Thornton, 1968*; *Jay et al., 2018*; *Palmer & Kronforst, 2020*). Such polymorphisms are rare in nature, with the most well studied examples occurring in papilionid butterflies (*Clarke, Sheppard & Thornton, 1968*; *Clarke & Sheppard, 1971*; *Hazel, 1990*; *Joron & Mallet, 1998*; *Nijhout, 2003*). For instance, polymorphisms in *Heliconious* species are determined by presence/absence of an introgressed chromosomal inversion 'supergene' (*Jay et al., 2018*), and alleles of a single ancestral gene (*doublesex*) control female-specific polymorphisms in *Papilio* species (*Palmer & Kronforst, 2020*).

In contrast to papilionid butterflies, the genetics of mimicry trait evolution among unionoid mussels is poorly understood. Unionoida comprise ~75% of the planet's freshwater bivalve species and are free-living apart from a brief, obligately parasitic, larval stage that infects fish hosts (*Bogan, 2007*; *Haag, 2012*). Gravid female mussels have evolved a spectrum of strategies to infect hosts with their larvae (*Zanatta & Murphy, 2006*; *Barnhart, Haag & Roston, 2008*; *Hewitt, Wood & Ó Foighil, 2019*). Females in many species use a mantle lure (*Welsh, 1933*): a pigmented fleshy extension that provides a visual cue resembling the prey of host fish, eliciting a feeding response that results in host infection (*Haag & Warren, 1999*; *Barnhart, Haag & Roston, 2008*; Fig. 1A). Many species also have a behavioral component; usually in the form of lateral undulations that travel as a wave along the edges of each half (right and left) of the mantle lure (*Ortmann, 1921*; *Barnhart, Haag & Roston, 2008*). Although this behavior was observed and described in the early 20th century (*Ortmann, 1921*), it was not until much later that *Haag & Warren (1999)* observed

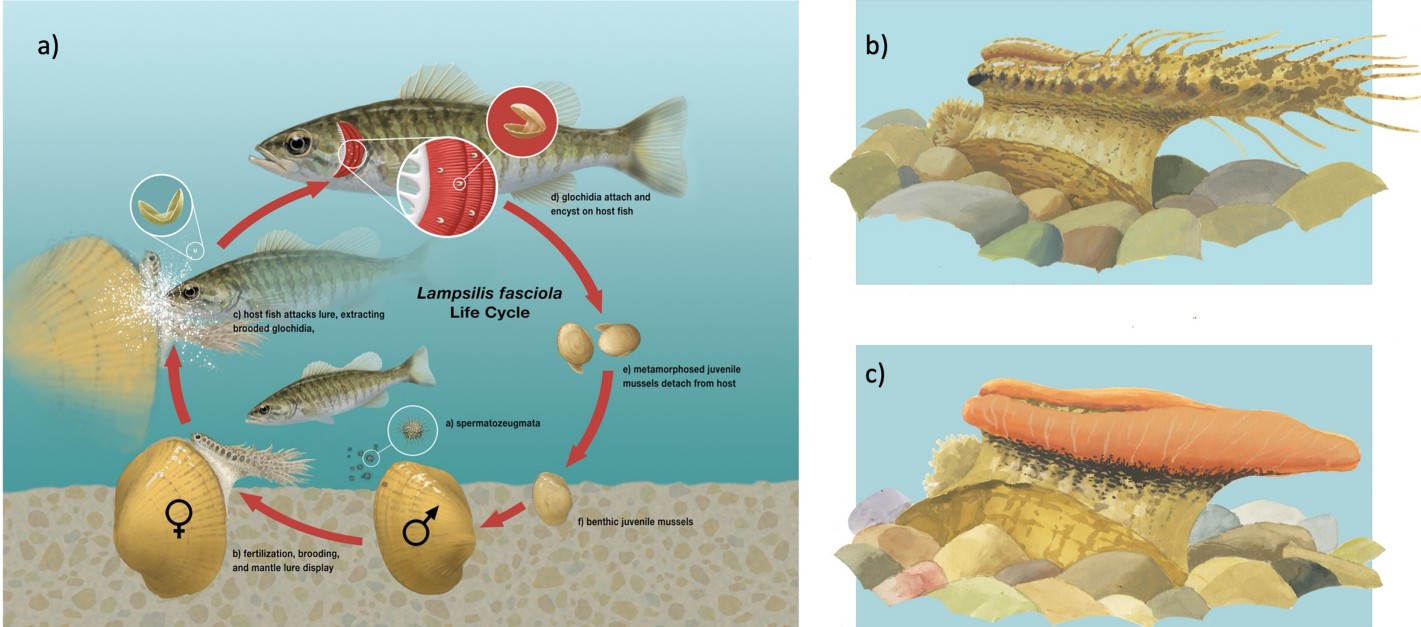

**Figure 1 Illustration of *Lampsilis fasciola* life cycle.** (A) The life cycle of the freshwater mussel *Lampsilis fasciola*. A gravid female mussel displays a mantle lure, here a darter mimic, to the primary fish host, *Micropterus dolomieu*. This elicits an attack through which the host is infected by mussel parasitic larvae (glochidia). After a short infective period (~2 weeks), the parasitic larvae metamorphose into juvenile mussels that detach from the host and fall to the substrate. (B) ("darter-like") and (C) ("worm-like") depict the two primary phenotypes of lure observed in *L. fasciola*. The former (B) has "eyespots", a mottled "main body" pigmentation composed of lateral and dorsal spots that can vary substantially in color, numerous and prominent marginal extensions, and a distinct "tail" region, whereas the latter lacks those features and has instead a uniform bright orange coloration underlain with a black basal stripe. Illustration by John Megahan.

how this behavior was used to attract strikes from host fish. The mantle lure presents itself as a reward to potential host fish but is deceptive in nature and leads to parasitization of the host fish. This mimetic system can therefore be classified as an example of aggressive mimicry following the definition by *Jamie (2017)*. The variability in lure display behavior among species of unionid is not well understood. Mimetic mantle lures predominate in Lampsilini, a major clade of North American freshwater mussels recently identified as a cryptic adaptive radiation centered on larval ecologies and specialized host-infection behaviors (*Hewitt, Haponski & Ó Foighil, 2021b*). This interaction is referred to as 'cryptic' because the specific host-parasite interactions are transient and difficult to determine *in-situ*. *Ortmann (1921)* and *Kramer (1970)* reported the production of rudimentary mantle lures in juveniles and male lampsilines, but noted that formation of fully developed lures is restricted to sexually mature females, and that only gravid females engage in lure display behaviors. Surprisingly, neither *Ortmann (1921)* nor *Kramer (1970)* depicted male mussel lure rudiments, nor could we find any such depictions in the literature.

Although mimetic mantle lures are a key adaptive trait of freshwater mussel diversification, the genetic regulators underlying their formation (*Kramer, 1970*), variation (*Haag, Warren & Shillingsford, 1999*; *Zanatta, Fraley & Murphy, 2007*; *Barnhart, Haag & Roston, 2008*), and evolution (*Zanatta & Murphy, 2006*; *Hewitt, Haponski & Ó Foighil, 2021b*) remain completely unknown. This gap in our knowledge is exacerbated by the stark

conservation status of North American freshwater mussels, with two thirds of species classified as threatened or near-threatened (*Lopes-Lima et al., 2018*).

As with papilionid butterflies (*Jay et al., 2018*; *Palmer & Kronforst, 2020*), targeting polymorphic lampsiline mantle lures for in-depth study may represent a tractable route to closing that gap between genes and phenotypes. *Lampsilis fasciola*, the Wavy-Rayed Lampmussel, is a promising candidate species in that it produces a number of distinct mantle lure phenotypes (*Zanatta, Fraley & Murphy, 2007*) across its Eastern North America distribution, extending from southern Ontario to northern Alabama (*Parmalee & Bogan, 1998*). Two range-wide lure phenotypes predominate in northern populations. The more common of the two, labeled "darter-like" by *Zanatta, Fraley & Murphy (2007)*, has "eyespots", a mottled "main body" pigmentation composed of lateral and dorsal spots that can vary substantially in color, numerous and prominent marginal extensions (AKA "appendages" or "tentacles"), and a distinct "tail" region (*Kramer, 1970*; *Zanatta, Fraley & Murphy, 2007*; Fig. 1B). A rarer lure phenotype, labeled "worm-like" by *McNichols (2007)*, lacks the above features and has instead a uniform bright orange coloration underlain with a black basal stripe (*Zanatta, Fraley & Murphy, 2007*; Fig. 1C). The latter lure phenotype is highly distinctive within the genus *Lampsilis* where fish-like mantle lures are the norm (*Kramer, 1970*). Much work has been done in attempt to quantify similarity between models and mimics, and qualitatively assess most likely models (*Kelly et al., 2021*), but defining models for lampsiline lure mimics thus far has largely been based on visual similarities defined by expert opinion (*Zanatta, Fraley & Murphy, 2007*; *Barnhart, Haag & Roston, 2008*). Based on the results of laboratory larval infection experiments and on the degree of ecological overlap, *Micropterus dolomieu* (Smallmouth Bass), and to a lesser extent *Micropterus salmoides* (Largemouth Bass), have been identified as *L. fasciola's* primary fish hosts (*Zale & Neves, 1982*; *McNichols, 2007*; *Morris et al., 2008*; *McNichols, Mackie & Ackerman, 2011*; *VanTassel et al., 2021*). Both host species are generalist predators of aquatic invertebrates and vertebrates (*Clady, 1974*).

Our study aimed to address outstanding, interrelated questions to develop *L. fasciola* into an integrated mantle lure polymorphism study system. First among them was residual uncertainty that the mantle lure morphs represent polymorphisms rather than cryptic species. *Zanatta, Fraley & Murphy (2007)*, using microsatellite markers, did not detect evidence of cryptic species but qualified their conclusions due to small sample sizes, and their result requires corroboration (*Fisheries & Oceans Canada, 2018*). Secondly, we currently lack any data on the mantle lure phenotype ratios over time (or on a mechanism for its presumed maintenance). Thirdly, we attempt to define respective models of each *L. fasciola* mantle lure mimic in a natural population. Finally, mantle lure display behavior is an important component of effective mimicry in freshwater mussels (*Welsh, 1933*; *Jansen, Bauer & Zahner-Meike, 2001*; *Haag & Warren, 2003*; *Barnhart, Haag & Roston, 2008*), but it is unknown if morphologically divergent *L. fasciola* mantle lures, that presumably mimic very distinct host prey models, also differ in their display behaviors. We tested this by making and analyzing video recordings of lure movements of displaying polymorphic females in a natural population over 3 years. We used a combination of field-collection, captive breeding, museum specimens, and ecological surveys to collect genetic,

phenotypic, and population data on this species. This publication was first released as a preprint (*Hewitt et al., 2023*; doi: https://doi.org/10.1101/2023.11.27.568842), however, the version presented here is the official peer-reviewed publication.

## MATERIALS AND METHODS

### Tissue sample collection

*L. fasciola* mantle tissue samples were collected for genotyping purposes by taking non-lethal mantle clip biopsies (*Berg et al., 1995*) from wild population lure-displaying female mussels during the summers of 2017, 2018, and 2021 in three rivers (Fig. 2). Maps were made in ArcGIS (*ESRI, 2022*) using *U.S. Geological Survey (2022)* as a basemap layer. Two of the sampling locations were in southeastern Michigan: the River Raisin at Sharon Mills County Park (42.176723, −84.092453; $N = 30$; 24 "darter-like", six "worm-like", collectively sampled in 2017, 2018 & 2020), and the Huron River at Hudson Mills Metropark, MI (42.37552, −83.91650; $N = 13$; 7 "darter-like", six "worm-like", collectively sampled in 2017, 2018, and 2020 under the MI Threatened and endangered species collection permit TE149). Both rivers flow into Lake Erie and are part of the Saint Lawrence drainage. The third location was in North Carolina: the Little Tennessee River ($N = 10$; 35.32324, −83.52275; $N = 10$, all were "darter-like" and sampled in 2017); this river is a tributary of the Tennessee River and part of the Mississippi drainage. Prior to each biopsy, photographs of the intact, undisturbed, lure display were taken with an Olympus Tough TG-6 underwater camera (Fig. S1).

### Captive brood tissue samples

We also obtained tissue samples from 50 captive-raised individuals of a single brood that had been ethanol-preserved. In 2009, the Alabama Aquatic Biodiversity Center (AABC) established a culture facility for endangered freshwater mussels. The Center's inaugural culture attempt, by co-authors Paul Johnson and Michael Buntin, was a proof-of-concept trial involving a single gravid female *L. fasciola* sourced from the Paint Rock River (another Tennessee River tributary; $N\,34°\,47.733′,W\,86°\,14.396′$) in Jackson County, AL (Fig. 2) on June 11, 2009. This female *L. fasciola* had a "worm-like" lure: the AABC data sheet for the trial 2009 host infection (Fig. S2) records that it was "bright orange and black" and lacked the "eyespots", mottled body coloration, marginal extensions, and "tail" of the "darter-like" lure phenotype (Buntin & Johnson, 2009, personal observations). On July 13 2009, about 31,000 glochidia larvae were extracted from the female's marsupia and used to infect *Micropterus coosae* (Redeye Bass) hosts sourced from the Eastaboga Fish Hatchery (Calhoun County, AL, USA) using standard protocols (*Barnhart, Haag & Roston, 2008*). The female mussel was then returned live to the Paint Rock River. Following completion of larval development on the fish hosts, about 9,300 metamorphosed juvenile mussels were recovered and reared, initially for the first few weeks in mucket bucket systems (*Barnhart, 2006*), then in a suspended upwelling system (SUPSYS) for 2 years with about 2,200 surviving. In 2011, this proof-of-concept culture experiment was terminated, and the survivors were donated to several research groups, with the majority used for toxicology experiments (*Leonard et al., 2014a, 2014b*).

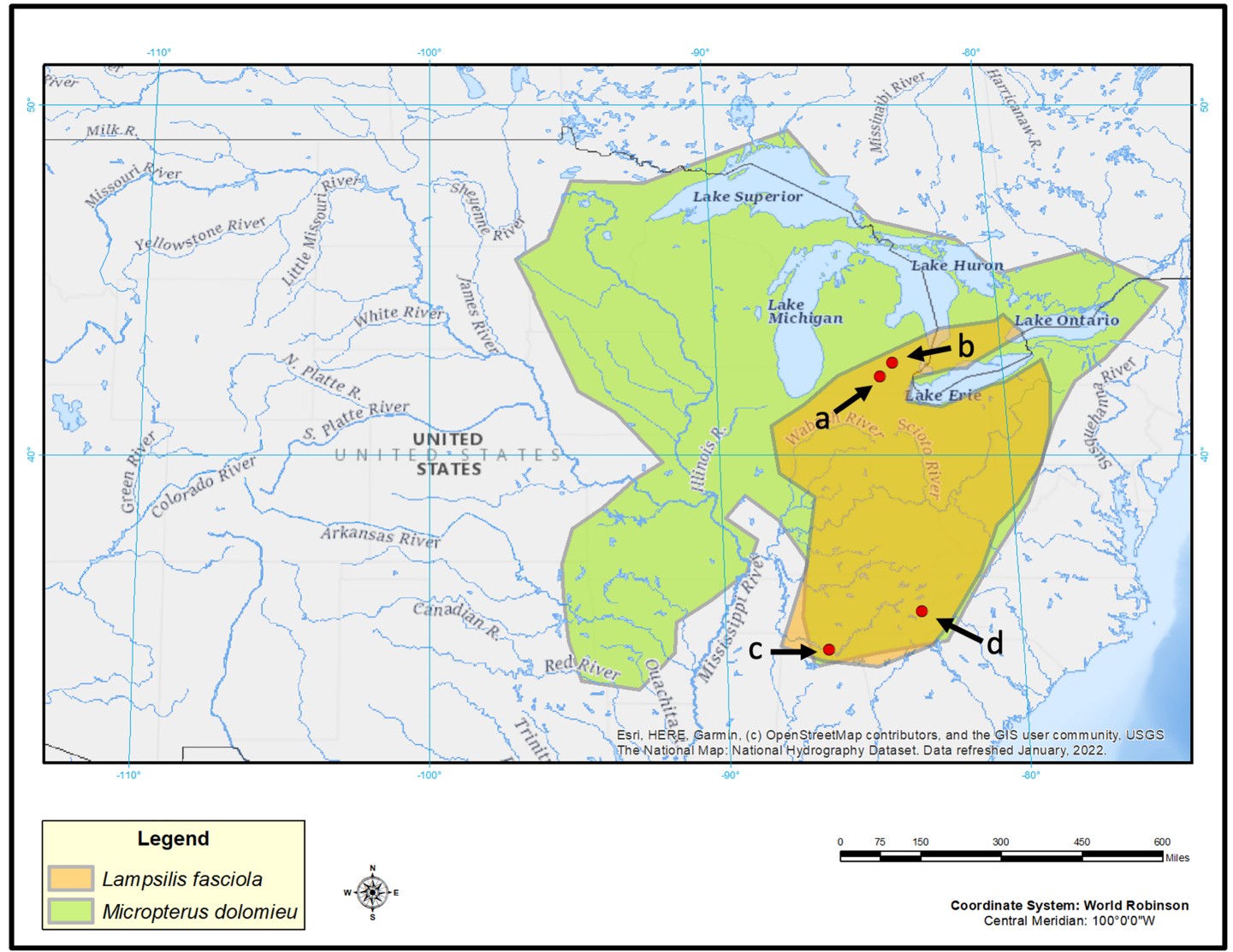

**Figure 2** **Map displaying geographic range of *Lampsilis fasciola* and its primary host, *Micropterus dolomieu*, as well as sampling locations.** Map of eastern North America showing the estimated ranges of *Lampsilis fasciola* (orange) and of its primary host fish *Micropterus dolomieu* (green). Red dots indicate sampling sites: Raisin River at Sharon Mils County Park (A), Huron River at Hudson Mills Park (B), Paint Rock River (C) and Little Tennessee River (D). Base map layer is from *U.S. Geological Survey (2022)*.

Prior to the brood's termination, Johnson noticed that a few females had attained sexual maturity and were displaying polymorphic lures (Figs. 3B, 3C). To substantiate that 2011 observation, we examined 50 individuals that had been preserved in 95% ethanol and shipped to Nathan Johnson (USGS) in Gainsville, FL in 2011. Because *Lampsilis* spp. juveniles and males produce a rudimentary mantle lure (*Ortmann, 1921*; *Kramer, 1970*), we were able to determine the primary lure phenotype (darter-like" or "worm-like") of all 50 preserved brood members. Using a Leica MZ16 dissecting microscope, individual photomicrographs were taken of the preserved rudimentary lure structures (Figs. 3D, 3E and S3), and their respective lure phenotypes were identified independently by both T.
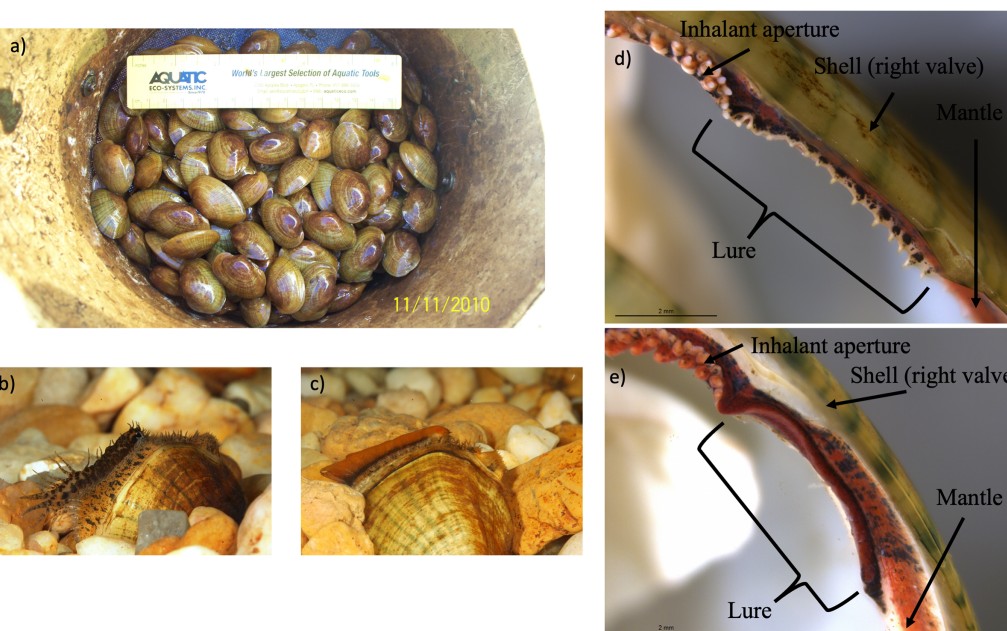

**Figure 3 Photographs of *Lampsilis fasciola* brood raised at the Alabama Aquatic Biodiversity Center, as well as photographs of preserved lure rudiments.** The *Lampsilis fasciola* brood raised at the Alabama Aquatic Biodiversity Center from a wild, gravid female, with a "worm-like" mantle lure, sampled from the Paint Rock River in June 2009. (A) shows juvenile members of the brood after ~16 months in culture. (B and C) show single, sexually maturing females after ~2 years of culture. The young female in (B) displayed a developing "darter-like" mantle lure (with "eyespots", mottled lateral coloration, marginal extensions, and a "tail") whereas her full- or half-sibling in (C) displayed a "worm-like" mantle lure (lacking the "darter" characteristics and having orange pigmentation with a black underlay). (D and E) Respectively show photomicrographs, taken with a dissecting microscope, of 95% ethanol-preserved rudimentary "darter-like" and "worm-like" lures from two additional brood members, part of a 50-individual subsample preserved in 2011.

Hewitt and by D. Ó Foighil. Additionally, tissue samples were acquired from all 50 individuals and included for phylogenomic analyses.

## Phylogenomic analyses

DNA sequencing and raw data processing were performed using the protocol outlined in *Hewitt, Haponski & Ó Foighil (2021a, 2021b)*. Genomic DNA was extracted from tissue samples using E.Z.N.A. Mollusk DNA kit (Omega Bio-Tek, Norcross, GA, USA) according to manufacturer's instructions and then stored at −80 °C. The quality and quantity of DNA extractions were assessed using a Qubit 2.0 Fluorometer (Life Technologies, Carlsbad, CA, USA) and ddRADseq libraries were prepared following the protocols of *Peterson et al. (2012)*. We then used 200 ng of DNA for each library prep. This involved digestion with Eco-RI-HF and MseI (New England Biolabs, Ipswich, MA, USA) restriction enzymes, followed by isolating 294–394 bp fragments using a Pippen Prep (Sage Science, Beverly, MA, USA) following the manufacturer's instructions. Prepared ddRADseq libraries then were submitted to the University of Michigan's DNA sequencing core and run in four different lanes using 150 bp paired-end sequencing on either an Illumina HiSeq 2500 or Illumina novaseq shared flow cell. Two control individuals of *L. fasciola* were run in each
lane and reads for both individuals clustered together in every analysis with 100% bootstrap support, indicating no lane effects on clustering among individuals. Raw demultiplexed data were deposited at GenBank under the bioproject ID PRJNA985631 with accession numbers SAMN35800743–SAMN35800847. Individuals included in phylogenomic analyses can be found in Table 1, and museum ID numbers can be found in Table S1.

The alignment-clustering algorithm in ipyrad v.0.7.17 (*Eaton, 2014*; *Eaton & Overcast, 2020*) was used to identify homologous ddRADseq tags. Ipyrad is capable of detecting insertions and deletions among homologous loci, which increases the number of loci recovered at deeper evolutionary scales compared to alternative methods of genomic clustering (*Eaton, 2014*). Demultiplexing was performed by sorting sequences by barcode, allowing for zero barcode mismatches (parameter 15 setting 0) and a maximum of five low-quality bases (parameter 9). Restriction sites, barcodes, and Illumina adapters were trimmed from the raw sequence reads (parameter 16 setting 2), and bases with low-quality scores (Phred-score < 20, parameter 10 setting 33) were replaced with an *N* designation. Sequences were discarded if they contained more than 5 N's (parameter 19). Reads were clustered and aligned within each sample at an 85% similarity threshold, and clusters with a depth <6 were discarded (parameters 11 and 12). We also varied the number of individuals required to share a locus from ~50% to ~75%.

We analyzed the two concatenated ddRAD-seq alignment files (50% and 75% minimum samples per locus) using maximum likelihood in RAxML v8.2.8 (*Stamatakis, 2014*). A general time-reversible model (*Lanave et al., 1984*) was used for these analyses that included invariable sites and assumed a gamma distribution. Support was determined for each node using 100 fast parametric bootstrap replications. Lure phenotype information was recorded and mapped on to the phylogenetic tree. Phylogenetic signal of lure phenotype was tested using *Pagel*'s *(1999)* $\lambda$ in R (*R Core Team, 2018*) with the 'phylobase' package (*Hackathon et al., 2013*).

## River raisin mantle lure phenotype ratios over time

Mid-20th century *L. fasciola* specimens collected at the Sharon Mills County Park site (Raisin River, MI, USA; Fig. 2A) are preserved as part of the University of Michigan's Museum of Zoology wet mollusk collection. They stem from eight different collecting events between 1954 and 1962 (Table S2), and their presence afforded an opportunity to assess the stability of the *L. fasciola* "darter/worm" mantle lure polymorphism in that population over a six-decade time interval. All of the museum specimens, males as well as females, were examined to determine whether their fully-formed (female) or rudimentary (male) mantle lures were "darter-like" or "worm-like". For females, this could be achieved by simple visual examination, but male lure classification required a dissecting microscope. The percentages of mantle lure phenotypes observed in the Sharon Mills County Park population was compared among mid-20th century (UMMZ preserved females and males) and 2017 (field photographs and videos of displaying females) samples using a Fisher's exact test, implemented in R.

**Table 1 The name, phenotype, and sequencing metrics.** Raw reads, total clusters, and total loci in assembly from the ddRAD sequencing are displayed for each genotyped sample of Lampsilis fasciola and of the outgroup taxa. Individual *Lampsilis fasciola* lure phenotype designation followed (*Zanatta, Fraley & Murphy, 2007*). Museum ID numbers can be found in Table S1.

| Sample name | Lure phenotype | Raw reads | Total clusters | Average clustering depth | Loci in assembly |
|---|---|---|---|---|---|
| L_fasciola_AL_brood_1 | Worm-like | 258,664 | 97,681 | 2.14 | 483 |
| L_fasciola_AL_brood_2 | Darter-like | 5,201,836 | 1,120,710 | 3.28 | 25,686 |
| L_fasciola_AL_brood_3 | Worm-like | 5,492,519 | 1,126,749 | 3.4 | 25,703 |
| L_fasciola_AL_brood_4 | Darter-like | 2,429,494 | 632,254 | 2.84 | 21,398 |
| L_fasciola_AL_brood_5 | Worm-like | 3,152,003 | 760,260 | 3.02 | 23,761 |
| L_fasciola_AL_brood_6 | Darter-like | 3,212,851 | 810,898 | 2.87 | 23,434 |
| L_fasciola_AL_brood_7 | Darter-like | 3,649,891 | 593,765 | 4.22 | 25363 |
| L_fasciola_AL_brood_8 | Darter-like | 4,869,307 | 1,462,723 | 2.29 | 19,089 |
| L_fasciola_AL_brood_9 | Worm-like | 3,158,818 | 718,169 | 3.08 | 23,033 |
| L_fasciola_AL_brood_10 | Darter-like | 4,000,321 | 915,881 | 3.12 | 24,916 |
| L_fasciola_AL_brood_11 | Worm-like | 5,679,854 | 1,171,842 | 3.35 | 25,770 |
| L_fasciola_AL_brood_12 | Darter-like | 4,212,783 | 979,265 | 3.04 | 24,693 |
| L_fasciola_AL_brood_13 | Worm-like | 1,300,563 | 399,134 | 2.51 | 12,145 |
| L_fasciola_AL_brood_14 | Darter-like | 4,100,372 | 1,043,360 | 2.79 | 23,521 |
| L_fasciola_AL_brood_15 | Darter-like | 5,804,293 | 1,412,102 | 2.91 | 25,570 |
| L_fasciola_AL_brood_16 | Worm-like | 1,555,906 | 427,061 | 2.7 | 14,099 |
| L_fasciola_AL_brood_17 | Darter-like | 2,073,968 | 598,680 | 2.59 | 13,668 |
| L_fasciola_AL_brood_18 | Worm-like | 6,919,783 | 1,574,429 | 3.08 | 25,811 |
| L_fasciola_AL_brood_19 | Darter-like | 3,434,210 | 829,507 | 2.94 | 23,708 |
| L_fasciola_AL_brood_20 | Darter-like | 4,778,853 | 994,416 | 3.35 | 25,500 |
| L_fasciola_AL_brood_21 | Worm-like | 2,462,560 | 590,095 | 2.91 | 20,588 |
| L_fasciola_AL_brood_22 | Worm-like | 6,600,876 | 1,406,451 | 3.26 | 26,080 |
| L_fasciola_AL_brood_23 | Darter-like | 7,090,859 | 1,628,965 | 3.06 | 25,932 |
| L_fasciola_AL_brood_24 | Worm-like | 4,546,435 | 1,061,394 | 3 | 24,174 |
| L_fasciola_AL_brood_25 | Worm-like | 5,379,577 | 1,135,906 | 3.35 | 25,703 |
| L_fasciola_AL_brood_26 | Worm-like | 5,592,652 | 1,501,130 | 2.67 | 23,965 |
| L_fasciola_AL_brood_27 | Worm-like | 4,893,957 | 825,855 | 4.09 | 25,924 |
| L_fasciola_AL_brood_28 | Darter-like | 2,596,873 | 519,103 | 3.59 | 22,103 |
| L_fasciola_AL_brood_29 | Darter-like | 3,401,334 | 883,485 | 2.87 | 21,377 |
| L_fasciola_AL_brood_30 | Worm-like | 3,876,395 | 1,014,133 | 2.8 | 22,072 |
| L_fasciola_AL_brood_31 | Worm-like | 5,391,442 | 1,246,528 | 3.07 | 25,009 |
| L_fasciola_AL_brood_32 | Darter-like | 4,365,005 | 1,084,596 | 2.85 | 23,030 |
| L_fasciola_AL_brood_33 | Darter-like | 5,116,507 | 1,117,916 | 3.16 | 24,667 |
| L_fasciola_AL_brood_34 | Darter-like | 7,480,755 | 1,601,100 | 3.19 | 26,163 |
| L_fasciola_AL_brood_35 | Darter-like | 8,121,426 | 1,825,135 | 3.02 | 25,972 |
| L_fasciola_AL_brood_36 | Darter-like | 5,521,997 | 1,414,238 | 2.78 | 24,163 |
| L_fasciola_AL_brood_37 | Darter-like | 6,562,641 | 1,579,514 | 2.88 | 25,476 |
| L_fasciola_AL_brood_38 | Darter-like | 6,303,766 | 1,596,624 | 2.76 | 24,448 |
| L_fasciola_AL_brood_39 | Darter-like | 6,206,795 | 1,488,925 | 2.91 | 24,648 |
| L_fasciola_AL_brood_40 | Darter-like | 8,630,897 | 1,891,164 | 3.11 | 26,176 |

(Continued)

| Sample name | Lure phenotype | Raw reads | Total clusters | Average clustering depth | Loci in assembly |
|---|---|---|---|---|---|
| L_fasciola_AL_brood_41 | Darter-like | 7,293,683 | 1,716,571 | 2.95 | 25,604 |
| L_fasciola_AL_brood_42 | Darter-like | 4,896,252 | 1,193,262 | 2.88 | 22,829 |
| L_fasciola_AL_brood_43 | Darter-like | 6,098,052 | 1,471,714 | 2.9 | 25,074 |
| L_fasciola_AL_brood_44 | Darter-like | 7,495,994 | 1,698,871 | 3.04 | 25,701 |
| L_fasciola_AL_brood_45 | Darter-like | 3,937,758 | 670,698 | 4.06 | 24,947 |
| L_fasciola_AL_brood_46 | Darter-like | 6,370,942 | 1,343,655 | 3.26 | 25,855 |
| L_fasciola_AL_brood_47 | Darter-like | 5,542,864 | 1,318,463 | 2.96 | 24,550 |
| L_fasciola_AL_brood_48 | Darter-like | 6,313,913 | 1,469,606 | 2.98 | 24,983 |
| L_fasciola_AL_brood_49 | Darter-like | 3,163,000 | 789,239 | 2.9 | 24,776 |
| L_fasciola_AL_brood_50 | Darter-like | 1,728,370 | 548,529 | 2.35 | 17,837 |
| L_fasciola_Huron_5 | Darter-like | 953,302 | 259,898 | 2.8 | 10,996 |
| L_fasciola_Huron_6 | Worm-like | 1,682,931 | 362,706 | 3.31 | 16,809 |
| L_fasciola_Huron_7 | Worm-like | 746,944 | 157,212 | 3.29 | 10,644 |
| L_fasciola_Huron_8 | Worm-like | 1,899,689 | 402,515 | 3.25 | 16,584 |
| L_fasciola_Huron_9 | Darter-like | 1,213,655 | 293,090 | 2.97 | 11,818 |
| L_fasciola_Huron_10 | Darter-like | 7,775,910 | 1,275,602 | 3.87 | 22,035 |
| L_fasciola_Huron_11 | Darter-like | 1,533,281 | 295,767 | 3.55 | 15,386 |
| L_fasciola_NC_1 | Darter-like | 1,308,813 | 254,002 | 3.61 | 11,873 |
| L_fasciola_NC_2 | Darter-like | 4,862,573 | 852,380 | 3.77 | 18,321 |
| L_fasciola_NC_3 | Darter-like | 663,874 | 165,869 | 2.95 | 9,960 |
| L_fasciola_NC_4 | Darter-like | 2,610,453 | 465,228 | 3.76 | 13,790 |
| L_fasciola_NC_5 | Darter-like | 6,927,947 | 1,459,334 | 3.05 | 20,804 |
| L_fasciola_NC_6 | Darter-like | 1,051,195 | 202,171 | 3.27 | 12,415 |
| L_fasciola_NC_7 | Darter-like | 1,948,092 | 382,878 | 3.61 | 17,101 |
| L_fasciola_NC_8 | Darter-like | 3,475,751 | 669,278 | 3.69 | 20,683 |
| L_fasciola_NC_9 | Darter-like | 5,693,936 | 1,634,946 | 2.46 | 22,325 |
| L_fasciola_NC_10 | Darter-like | 2,175,381 | 464,794 | 3.38 | 17,094 |
| L_fasciola_NC_11 | Darter-like | 2,189,933 | 516,643 | 3.05 | 17,580 |
| L_fasciola_Redo_1 | Darter-like | 1,455,864 | 327,622 | 2.62 | 13,478 |
| L_fasciola_Redo_2 | Darter-like | 1,839,020 | 436,418 | 2.43 | 13,181 |
| L_fasciola_Raisin_2 | Darter-like | 8,235,827 | 1,716,137 | 3.29 | 25,555 |
| L_fasciola_Raisin_3 | Darter-like | 6,032,935 | 1,488,448 | 2.85 | 25,006 |
| L_fasciola_Raisin_4 | Darter-like | 12,947,164 | 3,587,458 | 2.45 | 25,245 |
| L_fasciola_Raisin_1 | Darter-like | 6,639,384 | 1,086,218 | 3.97 | 23,458 |
| L_fasciola_Raisin_5 | Darter-like | 10,059,843 | 1,997,619 | 3.41 | 25,363 |
| L_fasciola_Raisin_6 | Darter-like | 8,019,689 | 1,847,955 | 3.01 | 25,769 |
| L_fasciola_Raisin_7 | Darter-like | 3,816,242 | 681,697 | 3.95 | 24,606 |
| L_fasciola_Raisin_8 | Darter-like | 6,117,037 | 1,282,299 | 3.27 | 22,439 |
| L_fasciola_Raisin_9 | Worm-like | 5,170,380 | 775,979 | 4.64 | 25,798 |
| L_fasciola_Raisin_10 | Darter-like | 761,451 | 176,858 | 3.14 | 11,477 |
| L_fasciola_Raisin_11 | Worm-like | 7,140,657 | 1,670,143 | 2.97 | 25,519 |

| Sample name | Lure phenotype | Raw reads | Total clusters | Average clustering depth | Loci in assembly |
|---|---|---|---|---|---|
| L_fasciola_Raisin_12 | Darter-like | 890,521 | 203,114 | 2.91 | 10,582 |
| L_fasciola_Raisin_13 | Darter-like | 1,071,361 | 225,030 | 3.47 | 13,512 |
| L_fasciola_Raisin_14 | Darter-like | 3,644,379 | 946,273 | 2.82 | 21,995 |
| L_fasciola_Raisin_15 | Darter-like | 3,578,043 | 482,446 | 5.04 | 17,514 |
| L_fasciola_Raisin_16 | Darter-like | 2,351,544 | 114,072 | 14.25 | 516 |
| L_fasciola_Raisin_17 | Darter-like | 5,272,816 | 1,304,726 | 2.87 | 23,305 |
| L_fasciola_Huron_1 | Worm-like | 13,366,692 | 4,050,829 | 2.26 | 17,555 |
| L_fasciola_Huron_2 | Darter-like | 2,819,896 | 928,226 | 2.24 | 20,205 |
| L_fasciola_Huron_3 | Darter-like | 662,275 | 186,602 | 2.66 | 7,653 |
| L_fasciola_Huron_4 | Darter-like | 4,792,093 | 855,457 | 3.88 | 24,512 |
| L_fasciola_AL_mom_1 | Darter-like | 8,095,030 | 1,840,917 | 2.95 | 25,420 |
| L_fasciola_AL_mom_2 | Darter-like | 10,329,331 | 3,504,027 | 2.03 | 24,488 |
| L_fasciola_AL_mom_3 | Darter-like | 10,384,477 | 2,987,559 | 2.34 | 25,056 |
| L_fasciola_Huron_12 | Worm-like | 6,906,349 | 1,672,394 | 2.87 | 25,281 |
| L_fasciola_Huron_13 | Worm-like | 6,955,496 | 1,670,627 | 2.88 | 25,593 |
| L_fasciola_Raisin_18 | Worm-like | 5,506,215 | 1,301,878 | 3 | 25,373 |
| L_fasciola_Raisin_19 | Worm-like | 6,611,596 | 1,524,682 | 3.03 | 25,604 |
| L_fasciola_Raisin_20 | Worm-like | 4,894,495 | 1,276,608 | 2.74 | 24,931 |
| L_fasciola_Raisin_21 | Worm-like | 8,396,562 | 1,736,736 | 3.26 | 25,490 |
| L_cardium_1 | | 6,864,226 | 1,710,220 | 2.8 | 14,625 |
| L_cardium_2 | | 4,898,330 | 1,091,622 | 3.11 | 13,433 |
| L_cardium_3 | | 7,109,883 | 2,005,565 | 2.5 | 14,563 |
| L_cardium_4 | | 4,637,077 | 997,208 | 3.27 | 13,860 |
| S_nasuta_1 | | 4,544,989 | 1,169,260 | 2.55 | 10,441 |

## Putative lure mimicry models

Population-specific putative model species for the *L. fasciola* mantle lure mimicry system were investigated at the River Raisin Sharon Mills County Park study site (Fig. 2), in part because of the availability of a comprehensive ecological survey of Raisin River fishes (*Smith, Taylor & Grimshaw, 1981*). "Darters"—members of the speciose North American subfamily Etheosomatinae— have been implictly identified as models for the predominant "darter-like" mantle lure phenotype (*Zanatta, Fraley & Murphy, 2007*), and they are preyed upon by *Micropterus dolomieu* (*Surber, 1941*; *Robertson & Winemiller, 2001*; *Murphy et al., 2005*), *L. fasciola's* primary fish host (*Zale & Neves, 1982*; *McNichols, 2007*; *Morris et al., 2008*; *McNichols, Mackie & Ackerman, 2011*; *VanTassel et al., 2021*). Ten species of Etheosomatinae occur in the River Raisin, as does *M. dolomieu* (*Smith, Taylor & Grimshaw, 1981*).

River Raisin gravid female *L. fasciola* engage in mantle lure displays from May-August. During the summer of 2017, a total of 27 different displaying females were photographed along a 150-m stretch downstream of the dam at Sharon Mills County Park using an

Olympus Tough TG-6 underwater camera. Individuals were located by carefully scanning the river bed with mask and snorkel to try and approximate the real ratios of phenotypes at this site. Additional lure photos were taken by coauthor Paul Johnson at the AABC of individuals from the Paint Rock River (AL). The lures were first categorized into broad groupings based on visual similarity, in terms of morphology and coloration. These groupings were then used to identify putative host prey fish model species from those present in the River Raisin drainage (Smith, Taylor & Grimshaw, 1981), based on similarities in size, shape, and coloration. Putative model species were further assessed based on their relative local abundance (Smith, Taylor & Grimshaw, 1981) and on their range overlap with both mimic and receiver. We also photograph and document the male rudimentary lures for both *L. fasciola* and *L. cardium*, taken from the River Raisin (Fig. S4). Geographic ranges of *L. fasciola*, the primary host *M. dolomieu*, and each prospective model species were produced by hand in Arcgis software (ESRI, 2022), and the overlap between *L. fasciola*, *M. dolomieu*, and each putative model species was assessed using Arcgis software.

## Behavioral analyses

Standardized video recordings of 27 mantle lure-displaying female *L. fasciola* (15 "darter-like" and 12 "leech-like") were recorded using a Go Pro Hero 6 camera in the summer of 2018 at the two different southeastern Michigan study sites: Sharon Mills County Park (River Raisin) and Hudson Mills Metropark (Huron River). All "darter-like" individuals were grouped together. An additional four video recordings of the lure behavior of sympatric *Lampsilis cardium*, a well studied congener lacking pronounced mantle lure polymorphisms (Kramer, 1970; Haag & Warren, 1999) were collected from the Sharon Mills site to assess interspecific variability in lure behavior. Recordings were captured from a top-down perspective during daylight hours using a standardized frame that included a metric ruler and a Casio TX watch to record date, time, and water temperature data within the video frame. For each displaying female, videos of the lure movements were recorded for 10 min at 120 frames-per-second. Setting up the camera occasionally disturbed the mussels, and video recordings began after waiting some time (usually 2–15 min) until the behavior qualitatively returned to its prior state. Analysis of the videos involved manually recording mantle lure movements for 20,000 frames (2.8 min), starting at 5,000 frames (42 s) to to avoid any camera shaking or hands accidentally blocking the view. The frame numbers when an individual movement began, defined as the first frame where contraction of mantle tissue was observed, and ended, defined as the time that mantle lure returns to it resting state, were noted. Movements of the left and the right mantle lure flaps were recorded seperately.

To quantitatively assess behavioral differences among samples, gait analysis diagrams were created in R for each displaying mussel. Because the lure is mimicking the swimming locomotion of fish, and fish locomotion has been characterized using gait analysis (Liao et al., 2003), we used gait analysis methods to characterize the non-locomotory motions that generate the luring behavior. Averages and standard deviations for the time intervals between lure undulations (the time between the start of one movement and the start of the
next) were calculated for each side of each individual, as well as duration undulation (the time between the start of one movement and the end of that movement) and proportion of movements synchronized. Movements were defined as synchronized if the start of a movement on one side was within four frames of the start of a movement on the corresponding side. Proportion of movements synchronized were calculated by dividing the number of synchronized movements by the sum of left movements only, right movements only, and synchronized movements. A Kruskal-Wallis test was used to test for overall differences among lure groups (*L. fasciola* "darter-like", *L. fasciola* "worm-like", and *L. cardium*), and pairwise Wilcoxon Signed rank tests were used to compare groups directly with a Bonferroni *p* value adjustment to correct for multiple tests. A Spearman correlation was used to test for an effect of water temperature on time interval between lure undulations.

To further explore differences in lure behavior among groups, we used a general linear mixed model (GLMM), with sample ID as a random factor, to test for differences in lure movement intervals. The GLMM approach, unlike simple mean comparisons, allows the inclusion of all lure movements for all individuals in the model. Because displaying mussels all varied in the number of lure movements recorded over 20,000 frames analyzed, a dataset of 1,000 random bootstrap values was constructed for each individual by randomly sampling values, with replacement. Models were fitted using the 'lmerTest' package in R, and *Satterthwaite*'s *(1946)* Method (*Kuznetsova, Brockhoff & Christensen, 2017*) was used to test for significance of fixed effects of lure phenotype on the interval between lure undulations.

## RESULTS

### Captive brood

Two independent classifiers concurred that the 50 preserved specimens from the same maternal brood included 33 "darter-like" (66%) and 17 "worm-like" (34%) individuals (Figs. 3D, 3E and S4).

### ddRAD-seq and phylogenomic analyses

Genomic sequencing returned raw reads ranging from 258,664 to 13,366,692 per individual across the 108 unionid specimens included in the analyses comprising samples of the ingroup *L. fasciola*, sourced from four different populations, along with outgroups *L. cardium* and *Sagittunio nasuta*. Mean coverage depth for the 85% clustering threshold ranged from 2.03 (L_fasciola_AL_mom_2) to 14.25 (L_fasciola_Raisin_16; Table 1). Between 28,725 and 16,161 homologous loci were identified across the two best ddrad datasets (85–50% and 85–75% respectively) and the number of loci recovered was generally consistent among all samples.

The maximum likelihood tree produced by RAxML (Fig. S4) recovered the following ingroup/outgroup topology: (*S. nasuta* (*L. cardium*, *L. fasciola*)) with outgroup branch lengths greatly exceeding those of the ingroup. To optimize the legibility of ingroup relationships, a compressed, color-coded graphic excluding *S. nasuta* was constructed (Fig. 4). A nested series of phylogenetic relationships was recovered for the four *L. fasciola*

fluvial populations with the two Michigan drainages being paraphyletic: (Little Tennessee River (Paint Rock River (River Raisin (River Raisin, Huron River)))). The ingroup topology also showed evidence of within-population genealogical relationships with all Paint Rock River brood members forming an exclusive clade (Fig. 4).

The respective primary mantle lure phenotypes—"darter-like" or "worm-like"—of all 92 *L. fasciola* ingroup individuals are indicated in Fig. 4. Note that three of the four population samples—Little Tennessee River, River Raisin and Huron River—were exclusively composed of mantle-lure displaying wild females, and the latter two samples were polymorphic in mantle lure composition. Regarding the Paint Rock River sample, polymorphic lures were restricted to the 50 captive-raised AABC brood members sourced from a gravid, wild female in 2009 (not included in the analyses). The ingroup phylogeny (Fig. 4) contained two polymorphic mantle lure clades, one composed of both Michigan populations (River Raisin and Huron River), the other consisting only of the AABC brood, and both clades had individuals of either lure phenotype interspersed across their respective topologies. Little phylogenetic signal associated with either primary mantle lure phenotype ($\lambda = 0.21$; $P = 0.13$).

## Phenotypic ratios over time

Table S2 summarizes the sex and primary lure phenotypes of 57 *L. fasciola* specimens collected from 1954–1962 at the River Raisin Sharon Mills County Park study site (Fig. 2A) and preserved in the University of Michigan Museum of Zoology's wet mollusk collection (Figs. 5B and 5C). These historical samples had a collective "darter-like" to "worm-like" ratio of 48:9, with 84.2% of individuals having the more common "darter-like" mantle lure phenotype and 15.8% having the "leech-like" phenotype. Figure 5A contrasts the mid-20[th] century lure phenotype ratios with a contemporary (2017) estimate in that same population, based on photographic recordings of 27 displaying females. The contemporary ratio was 23:4, with 85.2% of individuals having the more common "darter-like" mantle lure phenotype and 14.8% having the "leech-like" phenotype. The contemporary ratio was not significantly different from the historical ratio (Fisher Exact Test, $X^2 = 0.01$, $P = 0.91$).

## Putative raisin river lure mimicry models

The field photographs of 27 displaying female *L. fasciola* mantle lures in the Raisin River Sharon Mills County Park population in 2017 (Fig. S1) were categorized into either "darter-like" (*Zanatta, Fraley & Murphy, 2007*) or "worm-like" (*McNichols, 2007*), as summarized in the Materials & Methods section. In addition to the specific features that separate these two primary mantle lure phenotypes (presence/absence of "eyespots" mottled pigmentation, marginal extensions and a "tail"), "darter-like" lures exhibited a much higher degree of variation than did "worm-like" lures, both within populations and across the species range. The latter lure phenotype exhibited a relatively simple, uniform morphology combined with a bright orange coloration underlain with a black basal stripe phenotype in Michigan (Figs. 6F–6H), in Alabama (Figs. 6I and 6J), and in North Carolina populations (Fig. 2A in *Zanatta, Fraley & Murphy, 2007*). In contrast, Raisin River "darter-like" mantle lures exhibited individual-level variation that was sometime quite marked,

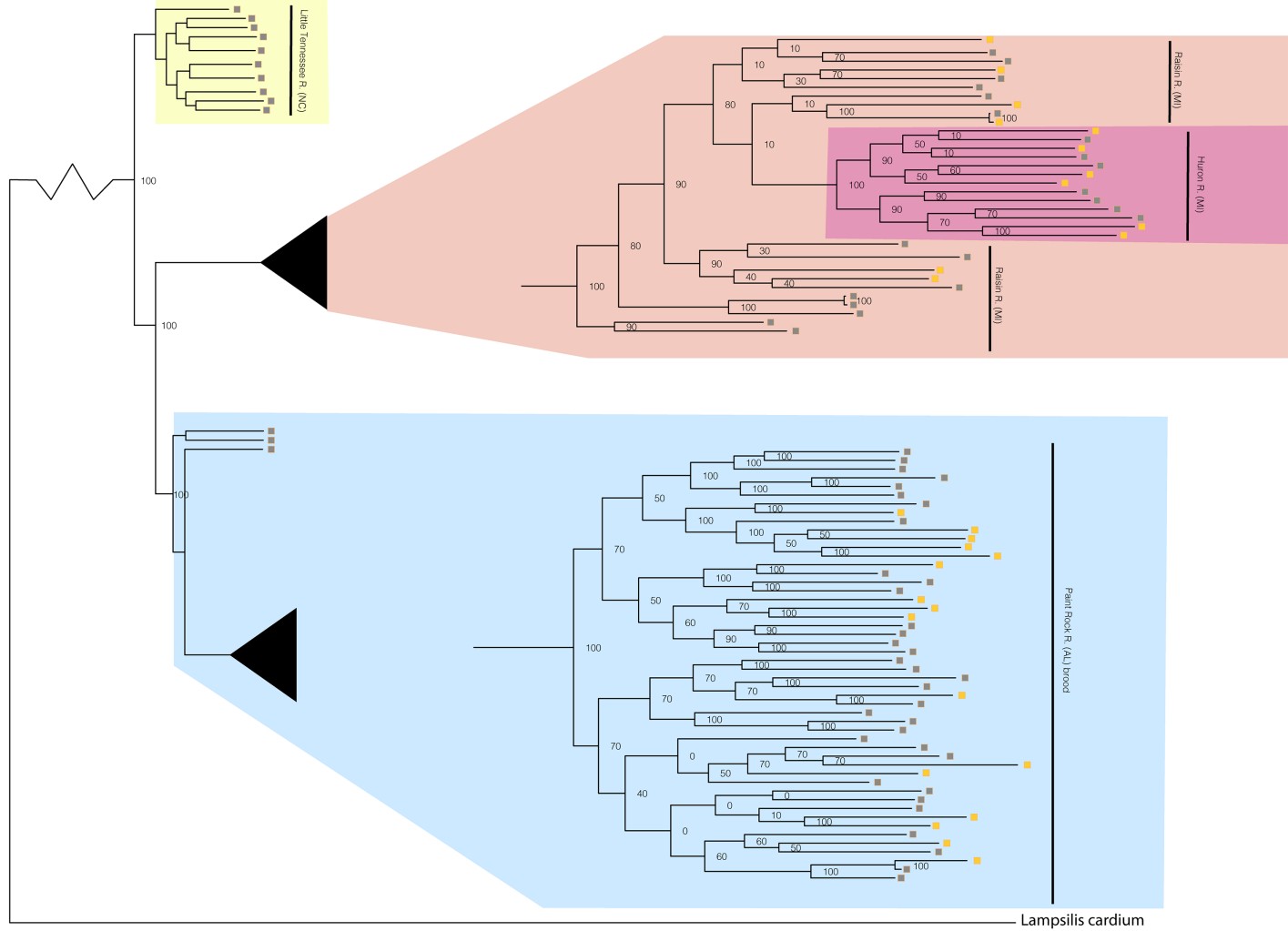

**Figure 4 Phylogenomic tree displaying *Lampsilis fasciola* from two MI populations, one NC population, and the Paint Rock River brood raised at the Alabama Aquatic Biodiversity Center.** Phylogenomic tree of 96 *Lampsilis fasciola* individuals created in RAxML using 28,735 concatenated ddRAD-seq loci. Gravid, lure-displaying females sampled from two Michigan drainages, River Raisin and Huron River, are respectively highlighted in peach and pink. Specimens sampled from the Paint Rock River, Alabama are highlighted in blue and consisted of three gravid, lure-displaying females, in addition to 50 larval brood members raised at the Alabama Aquatic Biodiversity Center in the zoomed-in tip clade. Gravid, lure-displaying females sampled from the Little Tennessee River in North Carolina are highlighted in yellow. Boxes indicate primary mantle lure phenotypes—"darter-like" (gray) or "worm-like" (orange)—of all *L. fasciola* individuals.     

especially in details of their pigmentation, and to a more limited degree in their marginal extensions (Figs. 6A–6D and S1). Among individual variation was most pronounced for inter-population camparisons, *e.g.*, see the much larger "tail" in the lure displaying Paint Rock River, Alabama specimen shown in Fig. 6E, and also the wider range of phenotypes present in North Carolina populations (Figs. 2B–2D in *Zanatta, Fraley & Murphy*'s *(2007)*. Male mantle lure rudiment photos are found in Fig. S5.

Despite the considerable individual variation among the 24 photographed Raisin River "darter-like" mantle lures (Fig. S1), it was possible to identify some shared phenotypic motifs, especially in pigmentation pattern, and to informally categorize 23/24 mantle lures

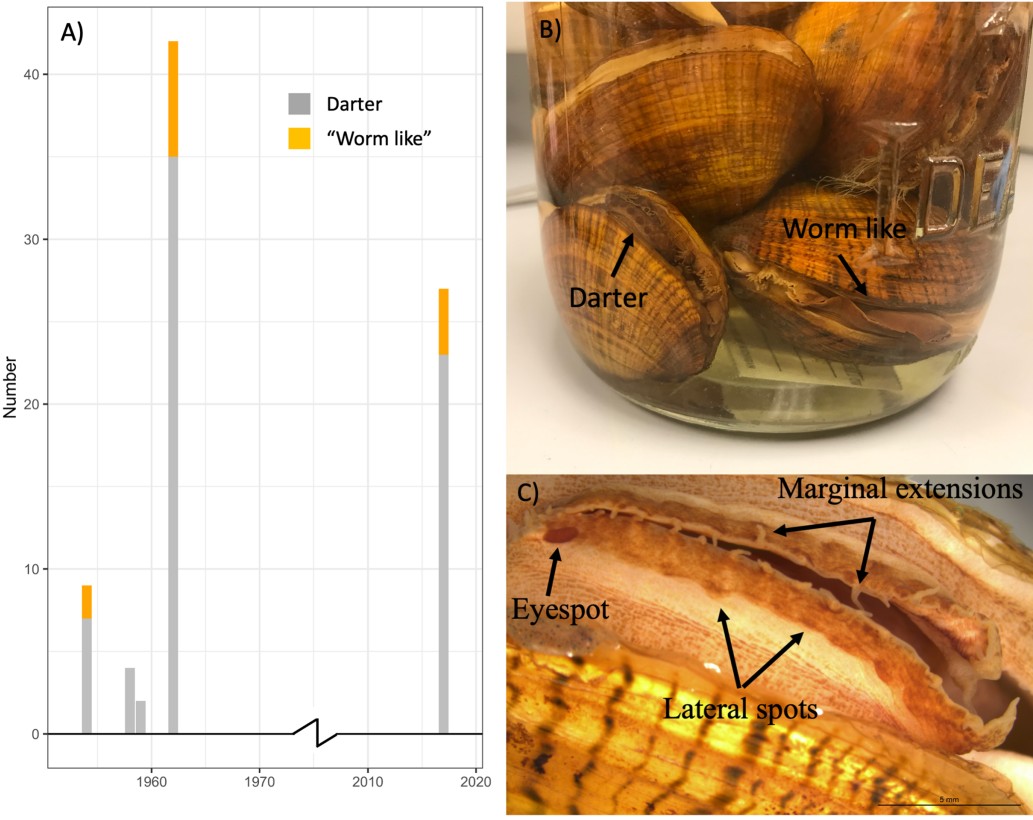

**Figure 5 The ratio of "worm-like" and "darter-like" *Lampsilis fasciola* lures over time in the River Raisin, MI, using historical and contemporary samples.** The observed frequency of River Raisin *Lampsilis fasciola* primary mantle lure phenotypes ("darter-like"; gray *vs.* "worm-like"; orange) at the Sharon Mills County Park study site during two different time periods. The 1954–1962 data were obtained from the University of Michigan Museum of Zoology (UMMZ) collection specimens, both female and male. The 2017 data were based on field observations of displaying females. (B) A jar of preserved UMMZ Sharon Mills specimens showing a "darter-like" and a "worm-like" mantle lure. (C) A "eyespot", lateral pigmented blotches, and marginal extensions in a "darter-like" lure of a preserved specimen.

with those shared motifs into four general groupings. Group 1 "darter-like" mantle lures were characterized by prominent, chevron-like, darker pigmented blotches, spaced regularly along the flanks of the lure, over a lighter background coloration (Fig. 6A). This general pattern occurred in 7/24 Raisin River "darter-like" lures examined. Group 2 was rarer (3/24 individuals) and consisted of a darker background coloration with large orange blotches spaced regularily along the lure flanks, some divided into "dorsal" and "ventral" elements (Fig. 6B). Group 3 (9/24 individuals) lures were characterized by prominent dark lateral maculation spatially divided into a "ventral" pattern of larger, regularly spaced blotches and a "dorsal" pattern of more numerous, irregular blotches of different sizes (Fig. 6C). Finally, Group 4 (3/24 individuals) lures were characterized by an evenly-dispersed, fine grained freckling of numerous pigmented spots over a lighter background (Fig. 6F).

To explore putative model species for the four *L. fasciola* Raisin River "darter-like" mantle lure groupings (Figs. 6A–6D and 6F), potential matches (in terms of size, shape and

### "Darter-like" Lure Variation

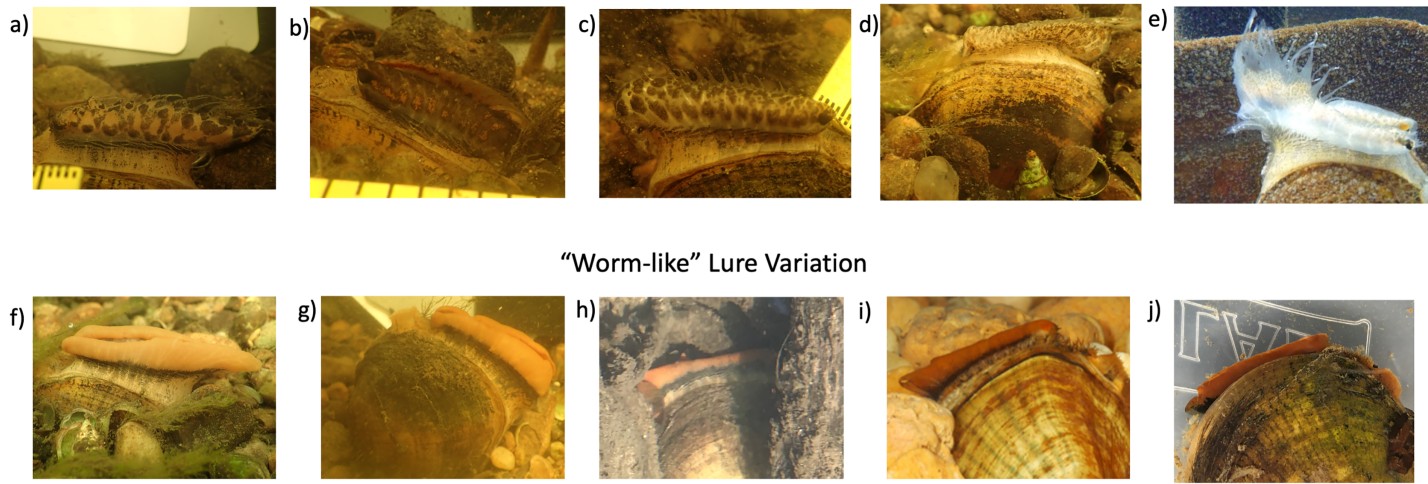

### "Worm-like" Lure Variation

**Figure 6** **Panel displaying variability in the two primary lure phenotypes of *Lampsilis fasciola*.** Variability in lure phenotype, both within a population and across the range of *Lampsilis fasciola*. (A–D) are "darter-like" Raisin River (MI) lures photographed in the field at Sharon Mills County Park. (E) Depicts a "darter-like" lure displayed by a Paint Rock River (AL) female. (F–H) show field photographs of "worm-like" lures displayed by three Sharon Mills females, with specimen H being a younger adult. (I and J) are photographs of two captive AABC specimens, with "worm-like" lures, sourced from the Paint Rock River. The former photo (I), taken in 2011, shows a young (2-year old) female, a member of the captive brood, displaying her lure, and the latter photo (J) is of a female field-sampled in 2022, and showing a partially retracted mantle lure.

coloration) were sought among the 10 species of Etheosomatidae that occur in the River Raisin (*Smith, Taylor & Grimshaw, 1981*), many of which display pronounced sexual dimorphism in body coloration (*Kuehne & Barbour, 2014*). The best apparent matches, depicted in Fig. 7, are as follows: Group 1 (Fig. 6A)-*Etheostoma blennioides* (female coloration), Group 2 (Fig. 6B)-*Etheostoma exile* (male coloration), Group 3 (Fig. 6C)-*Percina maculata* (male and female coloration) and Group 4 (Fig. 6D)-*Etheostoma microperca* (female coloration).

The distinctive color combination of the *L. fasciola* "worm-like" lure-solid orange with a black underlay (Figs. 6F–6J) does not match that of any Raisin River darter, or other Raisin River fishes (*Smith, Taylor & Grimshaw, 1981*). It does, however, match the coloration and size/shape, of the common North American leech, *Macrobdella decora*, which is widespread and abundant in eastern North America watersheds and typically feeds on aquatic vertebrates (*Klemm, 1982*; *Munro et al., 1992*). *M. dolomieu*, *L. fasciola*'s primary host fish, is a generalist predator with a diet of aquatic invertebrates, including leeches, in addition to small fishes (*Clady, 1974*), and recreational fishers frequently use live and/or artifical leeches as bait to catch this species (*Cooke et al., 2022*). Based on the available data, it seems that *Macrobdella decora* may be the best model species candidate for the "worm-like" (*McNichols, 2007*) *L. fasciola* mantle lure phenotype, and will hereafter be referred to as the leech phenotype.

The geographic range of the mimic, *L. fasciola*, is a subset of that of its receiver/host *M. dolomieu* (Fig. 2), and the extent of range overlap with all five putative River Raisin mantle lure models were calculated using Arcgis (Table 2) and are shown in Fig. 8. Three of

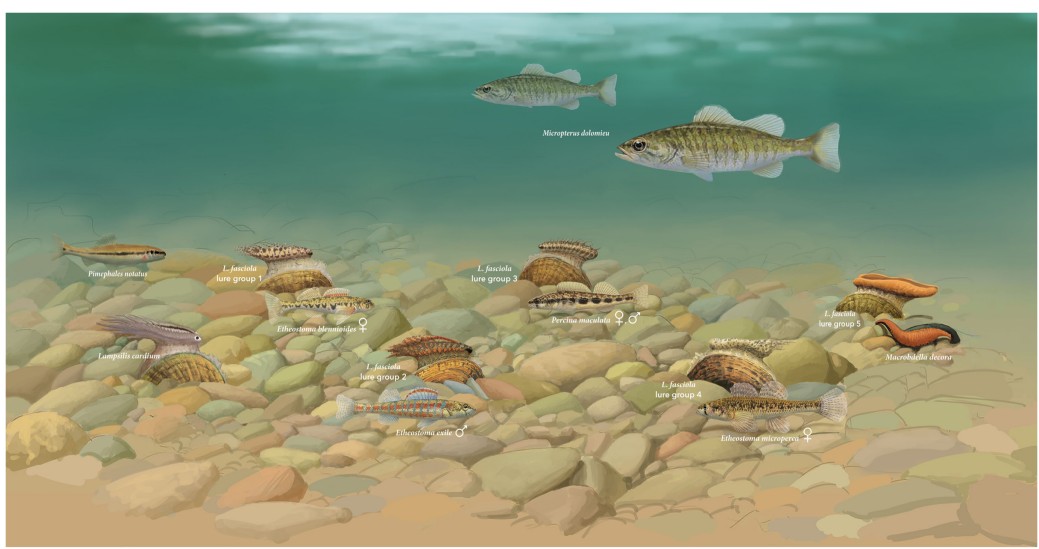

**Figure 7 Illustration of hypothetical benthic assemblage of main *Lampsilis fasciola* lure groups, and proposed models.** A hypothetical Raisin River (Michigan) benthic assemblage showing displaying exemplars of the putative five main *Lampsilis fasciola* mimetic mantle lure groups (Figs. 6A–6D and 6F) present at the Sharon Mills County Park study site, together with their respective model species, and their primary receiver/fish host, *Micropterus dolomieu*. Also shown is a displaying *Lampsilis cardium* with a "small minnow" mimetic mantle lure (*Patterson et al., 2018*) and its putative model, *Pimephales notatus*, the most common fish species in the River Raisin (*Smith, Taylor & Grimshaw, 1981*). Illustration by John Megahan.

**Table 2 Estimated range overlap between *Lampsilis fasciola* and five proposed models.** The five broad categories of lure phenotypes (Groups a–e) observed at the River Raisin Sharon Mills County Park population of *Lampsilis fasciola* (Fig. 2A), as well as the estimated geographic range overlap between *Lampsilis fasciola* and its five Raisin River putative model species.

| Type | Proposed model | Range overlap (km²) |
|---|---|---|
| Group a | *Etheostoma blennioides* | 480,731 |
| Group b | *Etheostoma exile* | 87,796 |
| Group c | *Percina maculata* | 525,772 |
| Group d | *Etheostoma microperca* | 164,539 |
| Group e | *Macrobella decora* | 419,259 |

the five putative models-*Etheostoma blenniodes*, *Percina maculata* and *Macrobdella decorata* have extensive overlap with *L. fasciola*'s range, but *E. exile* and *E. microperca* are restricted to northern portions.

## Behavioral analyses

Lure movements for both species consist of small undulations along the length of the mantle lure, beginning about two thirds of the way towards the "tail" side of the lure, and travelling towards the "head" of the lure. The *L. cardium* lure movements always occur on both left and right sides of the mantle lure simultaneosly, while both *L. fasciola* lure phenotyopes exhibit independent movement of the left and right sides of the lure.

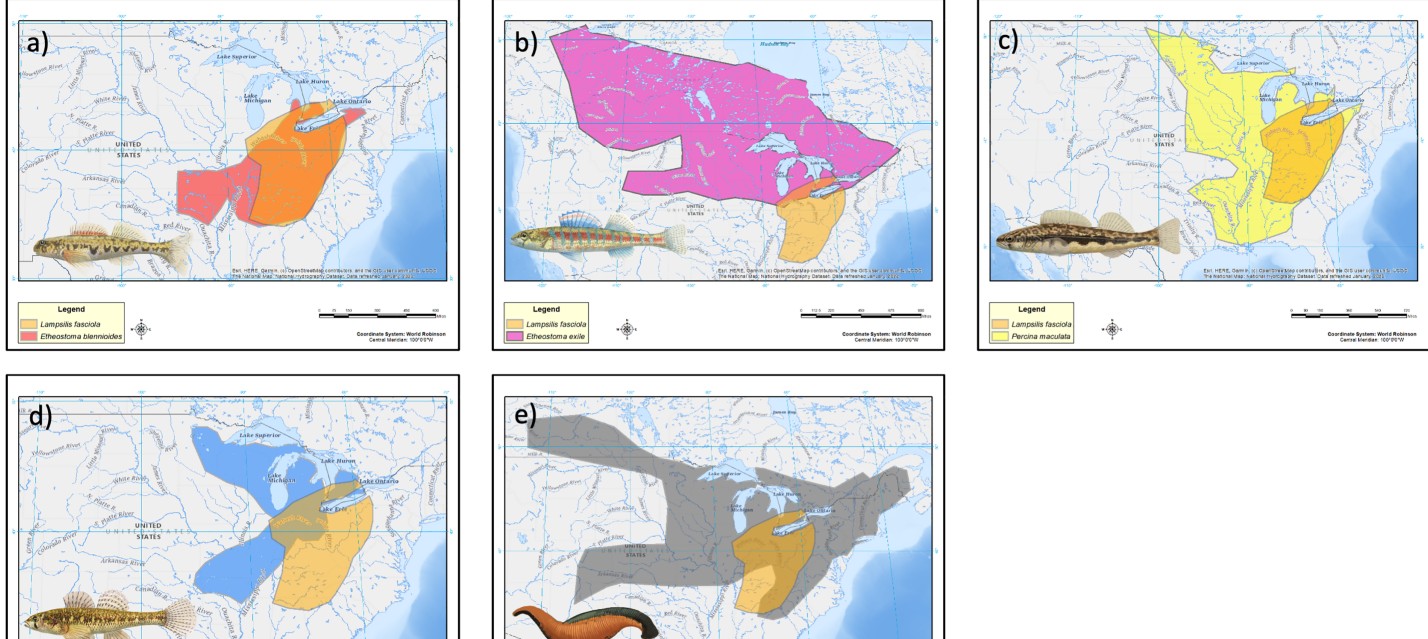

**Figure 8 Estimated range maps for proposed models of *Lampsilis fasciola* lures.** Estimated range maps for five proposed models for *Lampsilis fasciola* lures compared to the estimated geographic range of *Lampsilis fasciola* (orange). (A) *Etheostoma blennoides* (red), (B) *Etheostoma exile* (mauve), (C) *Percina maculata* (yellow), (D) *Etheostoma microperca* (blue), and (E) *Macrobdella decora* (gray). Note the differences in spatial scales in the panels. Model Illustrations by John Megahan. Base map layers are from *U.S. Geological Survey (2022)*.

Qualitatively, *L. fasciola* and *L. cardium* have very different mantle lure display behaviors. Gait diagrams show a clear distinction between *L. cardium* and both primary *L. fasciola* lure phenotypes ("darter" and "leech"). *L. cardium* consistently exhibited a synchronized lure undulation of both mantle lure flaps, whereas *L. fasciola* samples frequently moved left and right mantle flaps independently (Fig. 9 and S6). Gait diagrams also qualitatively showed that *L. fasciola* is charicterized by a high level of variability in undulation interval, *L. cardium* is much more regular in undulation interval with a steady beat frequency.

Intervals between movements in *L. cardium* were shorter (Wilcoxon test, W = 0, N = 4 *L. cardium*, 15 darter lure *L. fasciola*, 13 leech lure *L. fasciola, p* < 0.01 for both comparisons), less variable (Wilcoxon test, W = 0, *p* < 0.01 for both comparisons) and more synchronized (Wilcoxon test, W = 60, 48, *p* < 0.01 for comparisons with darter and leech lures, respectively) than in *L. fasciola* (Fig. 10). There was no difference in duration of lure undulations between *L. cardium* and both *L. fasciola* lure phenotypes (Wilcoxon test, W = 42,40, *p* = 0.26,0.06 for comparisons with darter and leech lures, respectively). Differences between the lure types of *L. fasciola* were smaller, with inter-movement intervals in the darter phenotype that were longer (Wilcoxon test, W = 142, *p* = 0.01) and marginally non-significantly more variable (W = 128, *p* = 0.07) but similar in duration (Wilcoxon test, W = 97, *p* = 0.76) and degree of synchronization (Wilcoxon test, W = 64,

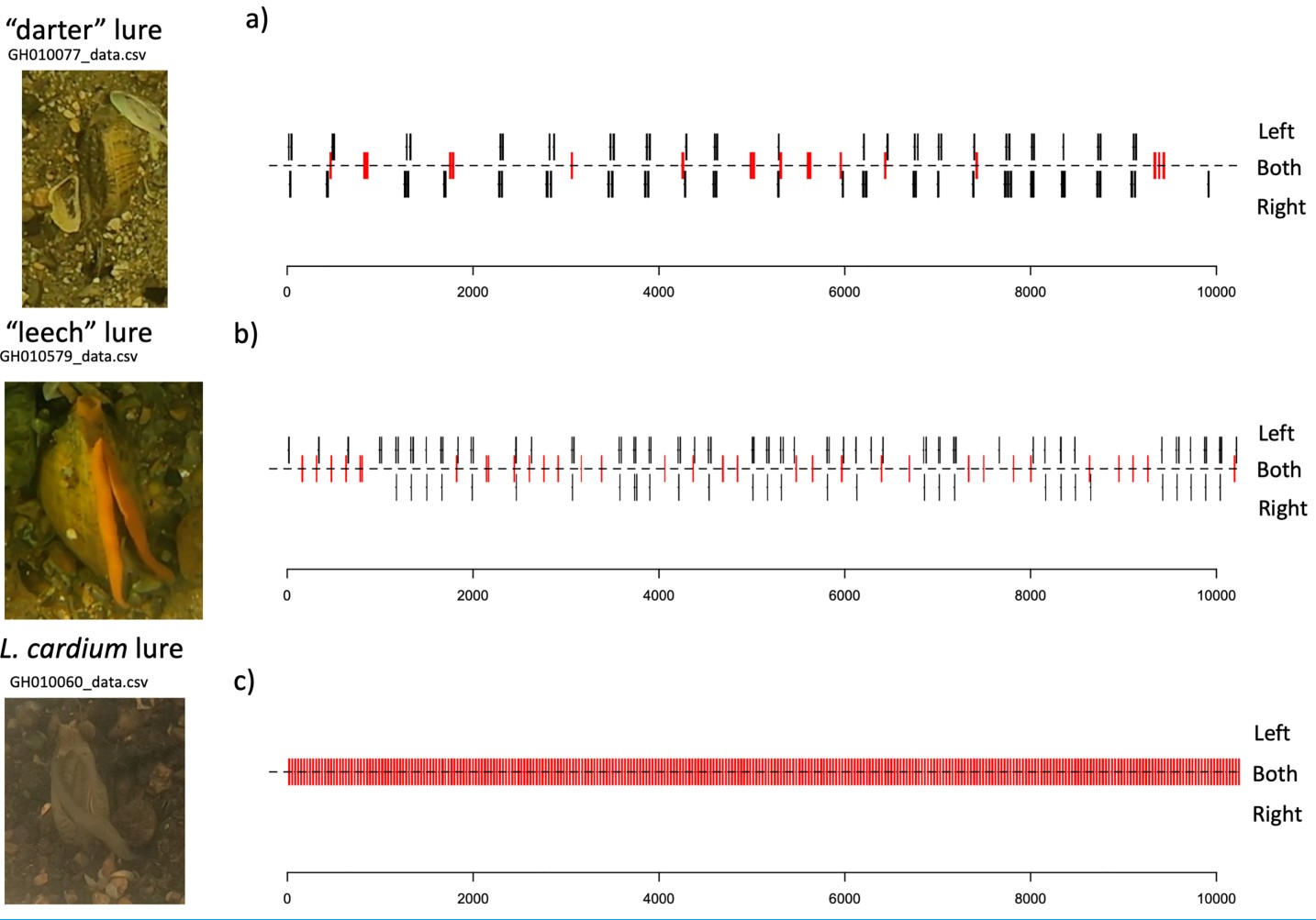

**Figure 9 Gait diagrams for three exemplar mussel displays; a "darter-like" _L. fasciola_, a "leech-like" _L. fasciola_, and a _Lampsilis cardium_.** Mantle lure gait diagrams for three representative individuals sampled. (A) shows a _Lampsilis fasciola_ "darter" lure sample (https://figshare.com/articles/media/ GH010077_cropped_mp4Polymorphism_in_the_aggressive_mimicry_lure_of_the_parasitic_freshwater_mussel_Lampsilis_fasciola/24850899), (B) displays a _Lampsilis fasciola_ "leech" lure sample (https://figshare.com/articles/media/GH010579_cropped_mp4Polymorphism_in_the_aggressive_ mimicry_lure_of_the_parasitic_freshwater_mussel_Lampsilis_fasciola/24850902), and (C) shows a _Lampsilis cardium_ sample (https://figshare.com/ articles/media/GH010060_cropped_mp4Polymorphism_in_the_aggressive_mimicry_lure_of_the_parasitic_freshwater_mussel_Lampsilis_fasciola/ 24847932). Red center lines indicate synchronized lure movement for both left and right mantle flaps, and black lines above and below the center line indicate independent left and right movements, respectively. The x-axis denotes time in seconds and frame number (120 fps).

_p_ = 0.22, Fig. 10). Table S3 details the time, date, location, temperature and summary statistics of all 34 lure display field recordings.

GLMM were used as an alternative analytical approach that included a large, bootstrapped dataset of lure movements. GLMM results were similar to those of the mean comparisons, with _L. cardium_ individuals having shorter movement intervals than either _L. fasciola_ lure morphs (an estimated 0.21 s for _L. cardium vs._ 3.2 and 1.0 s, respectively for _L. fasciola_ "darter" and "leech" lures). However, these fixed effects are not statistically significant.

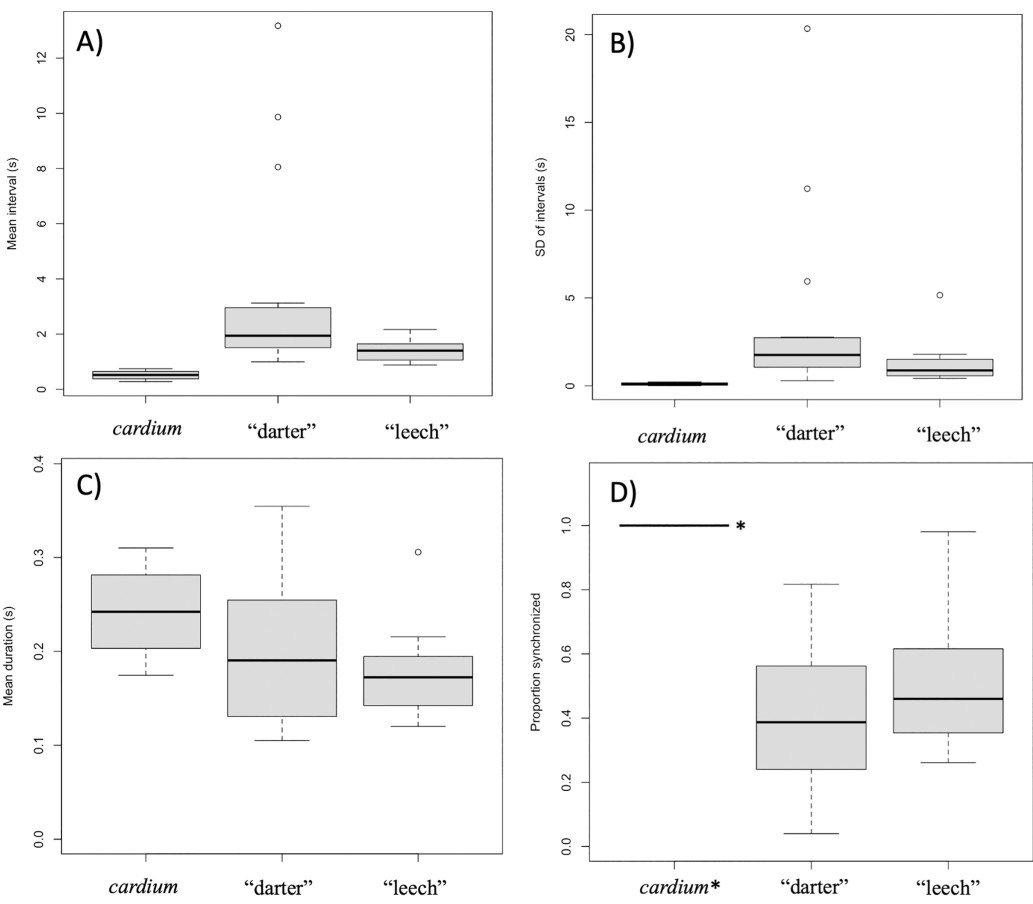

**Figure 10 Summary plots for behavioral analysis of the two primary *Lampsilis fasciola* lure phenotypes and *Lampsilis cardium*.** Boxplots from behavioral analyses of the two primary *Lampsilis fasciola* mantle lure phenotypes ("darter" *vs.* "leech", N = 15,12 respectively) and of *Lampsilis cardium* (N = 4). The middle line in the represents the median, the gray rectangle represents the interquartile range, the whiskers represent the minimum and maximum, excluding outliers, which are defined by 1.5* the interquartile range. (A) Comparison of the mean interval between movements of left mantle flap (s). (B) The standard deviation of lure movement interval (s) as a proxy for variability. (C) The average duration of each mantle lure movement. (D) The proportion of movements that are left-right synchronized. Note that the value for *L. cardium* is 1.0 (all lure movements for all individuals were synchronized) and indicated with *. (A–C) Show means for left mantle flap movements only.

## DISCUSSION

Two new pieces of evidence, phylogenomic and genetic, corroborated (*Zanatta, Fraley & Murphy*'s, *2007*) preliminary finding that the primary mantle lure morphs in *L. fasciola* (Figs. 1B and 1C) represent a within-population polymorphism rather than cryptic taxa. In phylogenomic analyses, all three polymorphic population samples (Huron, Raisin, and Paint Rock Rivers), collectively spanning the species range (Figs. 2A–2C), produced tip clades that were comprehensively polyphyletic regarding lure morph type (Fig. 4), and the "darter *vs.* leech" dichotomy yielded a low estimate of phylogenetic signal (λ = 0.21). However, the phylogenomic data did reveal clear evidence of geographic structuring (Fig. 4), with each geographic population forming discrete clades, even among regional

populations with a continuous freshwater connection. For example, the Huron and Raisin drainages empty in Western Lake Erie and the Little Tennese and Paint Rock drainages empty into the Tennessee River (see also *VanTassel et al. (2021)*). The Paint Rock River (AL) population was sister to the Michigan populations (Fig. 4), a result consistent with phylogeographic associations of multiple other North American species, including unionid mussels and *Micropterus dolomieu*, attributed to hypothesized glacial refugia in the southern Appalachian mountains (*Soltis et al., 2006*; *Borden & Krebs, 2009*; *Zanatta & Harris, 2013*; *Hewitt et al., 2018*).

Discovery of within-brood mantle lure heterogeneity (Fig. 3), apparently the first such record for unionids, confirms that the *L. fasciola* "darter-like" and "leech-like" mantle lures are polymorphisms rather then cryptic species, corroborating (*Zanatta, Fraley & Murphy, 2007*), and provides initial, although limited, genetic insights into lure phenotype inheritance. Of the 50 available offspring, the maternal "leech" phenotype was inherited by 17; the remaining 33 had the "darter" phenotype, but none exhibited a recombinant phenotype, *e.g.*, "leech" coloration with "darter" marginal extensions or "darter" coloration without marginal extensions. Evidence of discrete, within-brood segregation of the mantle lure polymorphism implies potential control by a single genetic locus and expression of the maternal phenotype in about one third of the offspring is inconsistent with a hypothetical dominant "leech" allele. Additional pedigree insights are currently inhibited by not knowing the number of sires that contributed to the brood: the dam was a wild-mated Paint Rock River individual. Freshwater mussel broods frequently have multiple paternity (*Ferguson et al., 2013*; *Wacker et al., 2018*). However, additional analyses of the RADseq dataset are needed to resolve that issue (*Thrasher et al., 2018*).

There are well-known cases of a single genetic locus controlling a mimic polymorphism in other systems. In butterflies, polymorphic mimicry in wing pigmentation is controlled by an introgressed mimicry supergene in *Heliconius* species (*Sheppard et al., 1985*; *Jay et al., 2018*) and by mimicry alleles of the transcription factor *doublesex* (*dsx*) in some *Papilio* species (*Palmer & Kronforst, 2020*). Note, however, that the *L. fasciola* mantle lure mimicry polymorphism differs in important ways from these butterfly systems. It is more complex because it involves putative models (darters and leeches) from disparate phyla rather than from similar morphospecies (other butterflies), thereby requiring polymorphic trait differentiation in pigmentation and in morphology (Figs. 1B and 1C). It is also a case of aggressive mimicry (*Jamie, 2017*), different from the Müllerian mimicry of *Heliconius* (*Kronforst & Papa, 2015*) or the Batesian mimicry of *Papilio* (*Kunte, 2009*).

Persistence of *L. fasciola* mantle lure polymorphism across a broad geographic scale (Fig. 2) is notable, although the mechanism responsible for widespread maintenace is unclear. One hypothesized mechanism for the persistence of polymorphisms in a species or population is frequency-dependent selection, where fitness is inversely proportional to frequency of a trait (*Clarke, 1964*; *Ayala & Campbell, 1974*). Frequency-dependent selection has been observed in other polymorphic mimicry systems (*Shine, Brown & Goiran, 2022*), and it has been suggested as a possible mechanism for persistence of the *L. fasciola* polymorphism (*Zanatta, Fraley & Murphy, 2007*; *Barnhart, Haag & Roston, 2008*; *Hewitt, Haponski & Ó Foighil, 2021b*). One criterion for frequency-dependent

selection is that phenotype ratios oscillate over time as initially rare phenotypes become more successful. However, the historical (1954–1962) and contemporary (2017) data from Sharon Mills County Park (Fig. 5) did not show evidence of such oscillation: the frequencies of the lures (darter lure = 84.2% *vs.* 85.2%, leech lure = 15.8% *vs.* 14.8%) remained essentially the same for both time windows, although we lack data for the intervening years. Theoretically, there are other mechanisms for balancing selection to maintain polymorphisms over long time-scales, including heterozygote advantage or opposing selection pressures favoring different alleles at polymorphic loci (*Ford, 1963*; *Prout, 2000*; *Mérot et al., 2020*), but underlying genetics of the *L. fasciola* polymorphism is unknown at this time, and more data are clearly needed.

The relative uniformity of the "leech" mantle lure phenotype in the River Raisin and throughout the *L. fasciola* range (Figs. 6F–6J) stands in sharp contrast with much higher local and range-wide variation shown by "darter" lures (Figs. 6A–6E). The four putative River Raisin darter model species–*Etheostoma blennioides*, *E. exile*, *E. microperca* and *Percina maculata*–are all common and widespread members of the drainage's ichthyofauna with 300–900 specimens of each species recovered from 30–100 sampling locations (out of 160 total) by the *Smith, Taylor & Grimshaw (1981)* ecological survey. That phenotypic lure disparity mirrors the collective phenotypic variability of darters *vs.* *Macrobdella decora*; darters are the second-most diverse fish clade in North America, with ~170 species (*Warren & Burr, 1994*; *Stein & Morse, 2000*). Another possibility is that at least some *L. fasciola* "darter-like" lures across the mussel's range are composite mimics of visual elements from more than one member of their local darter fauna. However, that remains to be established, as does the underlying nature of *L. fasciola* darter lure variation, *i.e.*, the degree to which it stems from a continuous spectrum of phenotypes or from the presence of additional discrete polymorphisms. The variability in "darter" lure phenotype does not seem to be associated with any environmental factors, which suggests this variability is not due to ecophenotypic plasticity, although more subtle factors, such as chemical cues, were not measured. Irrespective of the factors promoting variation among *L. fasciola* "darter" lure morphs, maintenance of close phenotypic tracking by lures of their respective models is expected, given host fishes' strongly adversive reactions to becoming infected (*Haag & Warren, 1999*).

While the behavior of mantle lures in *L.* mussels has been documented and studied for many decades (*Ortmann, 1921*; *Kramer, 1970*; *Haag & Warren, 1999*), detailed analysis of lure undulation behavior is currently lacking, and the relative importance of behavior *vs.* coloration and morphology is not well understood. The lure undulation for both *L. cardium* and *L. fasciola* starts about two thirds of the way to the "posterior" ("tailed") side of the lure, and then travels "forward" toward the "eyespot"-bearing "anterior". This is quite different from the oscilatory "S" shaped anterior-to-posterior swimming movements used by many fishes (*Liao, 2007*; *Smits, 2019*). However, it shares some resemblence to the "C" start behavior that many fishes use as an escape mechanism (*Witt, Wen & Lauder, 2015*). The unusual motion of the mantle lures may therefore be mimicking an escape behavior to some extent, but this remains to be established.

Although the *L. fasciola* behaviors differ significantly from those exhibited by *L. cardium*, there appears to be smaller behavioral polymorphism that distinguish the darter from leech lure phenotypes. Our putative model for River Raisin *L. cardium* mantle lures is a species of pelagic minnow, *Pimephales notatus* (Fig. 7), whose swimming behavior and ecology differs markedly from that of darters (*Burress et al., 2017*). Darters have lost or greatly reduced their swim bladder and are primarily benthic in habit, spending much of their time resting on the stream bed with slight body movements caused by ambient water flow (*Demski, Gerald & Popper, 1973*; *Zeyl et al., 2016*). They intermittently swim by "hopping" across the substrate using pectoral fins and caudal undulations in a manner that is much more erratic than the midwater swimming behavior of most minnows (*Winn, 1958*; personal observations). This matches a general difference observed between *L. cardium* and *L. fasciola* lures: *L. cardium* lures move faster and more regularly in a highly synchronized way, in contrast with the erratic, often left-right-unsynchronized movements of *L. fasciola* lures, apart from slight passive undulations caused by the ambient river currents. Unfortunately, the sample size of *L. cardium* was low ($N = 4$), despite a great deal of effort, trying to locate gravid female *L. cardium* that were actively displaying.

The only major difference in lure behavior between the "darter" and "leech" lure behvior of *L. fasciola* is a slightly slower rate exhibited by the "darter" lures, and marginally non-significant differences in variability between lure undulations. Both *L. fasciola* morphs have a similar erratic motion, despite the polymorphism putatively modeling taxa from disparate phyla. Leeches swim by a dorsoventeral bending wave moving from head to tail (*Jordan, 1998*). This swimming behavior is very different from the lure undulations observed in the leech-like *L. fasciola* lures. It is possible that leech behavior differs when moving along the substrate, where displaying *L. fasciola* are located, but we currently lack data on leech swimming behavior in different environments. The ecological importance of the minor, but statistically significant, differences in overall lure beat frequency observed between "darter" and "leech" mimics (Fig. 10) is difficult to evaluate at present, and it remains to be established if it, like the lure morphological differences, is also under genetic control. One additional caveat is that we focused primarily on differences in the timing of mantle lure displays, which were the most practical to measure *in-situ* with the ambient river flow. We also did not have any data on possible chemosensory cues that could potentially be involved.

Our discovery of discrete within-brood inheritance of the *L. fasciola* lure polymorphism is of particular interest because it implies potential control by a single genetic locus. There are a number of parallel cases in the recent literature, *e.g.*, in butterflies, the regulation of polymorphic mimicry in wing pigmentation also involves single genetic loci (*Jay et al., 2018*; *Palmer & Kronforst, 2020*). *Timmermans et al. (2020)* used SNP data from *Papilio dardanus* to discover a genomic inversion associated with its mimetic polymorphism, and this approach is likely also tractable for *L. fasciola* given the occurance of polymorphic brood. We are currently raising an additional polymorphic brood at the AABC. Mantle lures are a key adaptive trait in Lampsiline evolution and diversification (*Hewitt, Haponski & Ó Foighil, 2021b*), and *L. fasciola* is a promising and highly tractable

model system to uncover the genetics of lure development and variation in a unionoid mussel.

## ACKNOWLEDGEMENTS

We thank our reviewers and the handling editor, Donald Kramer, for the thorough review of this manuscript. We also would like to thank University of Michigan Museum of Zoology artist John Megahan for providing original artwork for Figs. 1, 7, and 8. Finally, we would like to acknowledge support from the University of Michigan Museum of Zoology and from the John B. Burch Malacology fund.

### Funding

This work was financially supported through block grant funding through Rackham School of Graduate Studies, the Dr. Nancy Williams Walls award for field research, and the State Wildlife Grant Program. The funders had no role in study design, data collection and analysis, decision to publish, or preparation of the manuscript.

### Grant Disclosures

The following grant information was disclosed by the authors:
Rackham School of Graduate Studies.
Dr. Nancy Williams Walls award.
State Wildlife Grant Program.

### Competing Interests

The authors declare that they have no competing interests.

### Author Contributions

- Trevor L. Hewitt conceived and designed the experiments, performed the experiments, analyzed the data, prepared figures and/or tables, authored or reviewed drafts of the article, and approved the final draft.
- Paul D. Johnson performed the experiments, authored or reviewed drafts of the article, and approved the final draft.
- Michael Buntin performed the experiments, authored or reviewed drafts of the article, and approved the final draft.
- Talia Y. Moore conceived and designed the experiments, authored or reviewed drafts of the article, and approved the final draft.
- Diarmaid Ó Foighil conceived and designed the experiments, authored or reviewed drafts of the article, and approved the final draft.

### Field Study Permissions

The following information was supplied relating to field study approvals (*i.e.*, approving body and any reference numbers):

Tissue samples from the field were collected under the Michigan Threatened and Endangered Species Collection Permit #TE149.

## DNA Deposition

The following information was supplied regarding the deposition of DNA sequences:

The raw demultiplexed ddRAD sequences are available at GenBank: SAMN35800743–SAMN35800847.

## Data Availability

The code used to analyze behavioral data, create figures, and run the statistical models is available in the Supplemental File.

## Supplemental Information

Supplemental information for this article can be found online at http://dx.doi.org/10.7717/peerj.17359#supplemental-information.

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

# PeerJ

**Hewitt TL, Haponski AE, Ó Foighil D. 2021a.** *ddRAD-seq alignment data for unionid mussels [Dataset].* Dryad DOI 10.5061/dryad.c866t1g62.

**Hewitt TL, Haponski AE, Ó Foighil D. 2021b.** Evolution of diverse host infection mechanisms delineates an adaptive radiation of lampsiline freshwater mussels centered on their larval ecology. *PeerJ* **9**:e12287 DOI 10.7717/peerj.12287.

**Hewitt TL, Johnson P, Buntin M, Moore T, Ó Foighil D. 2023.** Aggressive mimicry lure polymorphisms in the parasitic mussel *Lampsilis fasciola* model fish or leech host prey and differ in morphology and pigmentation, but not in display behavior. BioRxiv preprints DOI 10.1101/2023.11.27.568842.

**Hewitt TL, Wood CL, Ó Foighil D. 2019.** Ecological correlates and phylogenetic signal of host use in North American unionid mussels. *International Journal for Parasitology* **49(1)**:71–81 DOI 10.1016/j.ijpara.2018.09.006.

**Jamie GA. 2017.** Signals, cues and the nature of mimicry. *Proceedings of the Royal Society B: Biological Sciences* **284(1849)**:20162080 DOI 10.1098/rspb.2016.2080.

**Jansen W, Bauer G, Zahner-Meike E. 2001.** Glochidial mortality in freshwater mussels. In: Bauer G, Wächtler K, eds. *Ecology and Evolution of the Freshwater Mussels Unionoida.* Berlin, Heidelberg: Springer Berlin Heidelberg, 185–211 DOI 10.1007/978-3-642-56869-5_11.

**Jay P, Whibley A, Frézal L, Rodríguez de Cara MÁ, Nowell RW, Mallet J, Dasmahapatra KK, Joron M. 2018.** Supergene evolution triggered by the introgression of a chromosomal inversion. *Current Biology* **28(11)**:1839–1845.e3 DOI 10.1016/j.cub.2018.04.072.

**Jordan CE. 1998.** Scale effects in the kinematics and dynamics of swimming leeches. *Canadian Journal of Zoology* **76(10)**:1869–1877 DOI 10.1139/z98-131.

**Joron M, Mallet JLB. 1998.** Diversity in mimicry: paradox or paradigm? *Trends in Ecology & Evolution* **13(11)**:461–466 DOI 10.1016/S0169-5347(98)01483-9.

**Kelly MBJ, McLean DJ, Wild ZK, Herberstein ME. 2021.** Measuring mimicry: methods for quantifying visual similarity. *Animal Behaviour* **178(11)**:115–126 DOI 10.1016/j.anbehav.2021.06.011.

**Klemm DJ. 1982.** *Leeches (Annelida: Hirudinea) of North America.* Cincinnati, Ohio: U. S. Environmental Protection Agency.

**Kramer L. 1970.** The mantle flap in three species of *Lampsilis. Malacologia* **10**:225–282 DOI 10.7302/14751.

**Kronforst MR, Papa R. 2015.** The functional basis of wing patterning in *Heliconius* butterflies: the molecules behind mimicry. *Genetics* **200**:1–19 DOI 10.1534/genetics.114.172387.

**Kuehne RA, Barbour RW. 2014.** *The American darters.* University press of Kentucky.

**Kunte K. 2009.** The diversity and evolution of Batesian mimicry in *Papilio* swallowtail butterflies. *Evolution: International Journal of Organic Evolution* **63**:2707–2716 DOI 10.1111/j.1558-5646.2009.00752.x.

**Kuznetsova A, Brockhoff PB, Christensen RHB. 2017.** lmerTest package: tests in linear mixed effects models. *Journal of Statistical Software* **82(13)**:1–26 DOI 10.18637/jss.v082.i13.

**Lanave C, Preparata G, Sacone C, Serio G. 1984.** A new method for calculating evolutionary substitution rates. *Journal of Molecular Evolution* **20(1)**:86–93 DOI 10.1007/BF02101990.

**Leonard JA, Cope WG, Barnhart MC, Bringolf RB. 2014a.** Metabolomic, behavioral, and reproductive effects of the aromatase inhibitor fadrozole hydrochloride on the unionid mussel *Lampsilis fasciola. General and Comparative Endocrinology* **206**:213–226 DOI 10.1016/j.ygcen.2014.07.019.

**Leonard JA, Cope WG, Barnhart MC, Bringolf RB. 2014b.** Metabolomic, behavioral, and reproductive effects of the synthetic estrogen 17 α-ethinylestradiol on the unionid mussel *Lampsilis fasciola*. *Aquatic Toxicology* **150(Suppl. 1)**:103–116 DOI 10.1016/j.aquatox.2014.03.004.

**Liao JC. 2007.** A review of fish swimming mechanics and behaviour in altered flows. *Philosophical Transactions of the Royal Society B: Biological Sciences* **362(1487)**:1973–1993 DOI 10.1098/rstb.2007.2082.

**Liao JC, Beal DN, Lauder GV, Triantafyllou MS. 2003.** The Kármán gait: novel body kinematics of rainbow trout swimming in a vortex street. *Journal of Experimental Biology* **206(6)**:1059–1073 DOI 10.1242/jeb.00209.

**Lopes-Lima M, Burlakova LE, Karatayev AY, Mehler K, Seddon M, Sousa R. 2018.** Conservation of freshwater bivalves at the global scale: diversity, threats and research needs. *Hydrobiologia* **810(1)**:1–14 DOI 10.1007/s10750-017-3486-7.

**Maran T. 2015.** Scaffolding and mimicry: a semiotic view of the evolutionary dynamics of mimicry systems. *Biosemiotics* **8(2)**:211–222 DOI 10.1007/s12304-014-9223-y.

**McNichols KA. 2007.** Implementing recovery strategies for mussel species at risk in Ontario. Dissertation, University of Guelph.

**McNichols KA, Mackie GL, Ackerman JD. 2011.** Host fish quality may explain the status of endangered *Epioblasma torulosa rangiana* and *Lampsilis fasciola* (Bivalvia Unionidae) in Canada. *Journal of the North American Benthological Society* **30**:60–70 DOI 10.1899/10-063.1.

**Morris TJ, McGoldrick DJ, Metcalfe-Smith JL, Zanatta DT, Gillis P. 2008.** *Pre-COSEWIC assessment of the federally endangered Wavyrayed Lampmussel (Lampsilis fasciola)*. Burlington, ON: DFO Canadian Science Advisory Secretariat.

**Munro R, Siddall M, Desser SS, Sawyer RT. 1992.** The leech as a tool for studying comparative haematology. *Comparative Haematology International* **2**:75–78 DOI 10.1007/BF00186263.

**Murphy GW, Newcomb TJ, Orth DJ, Reeser SJ. 2005.** Food habits of selected fish species in the shenandoah river Basin, Virginia. In: *Proceedings of the Annual Conference of the Southeastern Association of Fish and Wildlife Agencies*, Vol. 59325–335.

**Mérot C, Llaurens V, Normandeau E, Bernatchez L, Wellenreuther M. 2020.** Balancing selection via life-history trade-offs maintains an inversion polymorphism in a seaweed fly. *Nature Communications* **11**:1–11 DOI 10.1038/s41467-020-14479-7.

**Nijhout HF. 2003.** Polymorphic mimicry in Papilio dardanus: mosaic dominance, big effects, and origins. *Evolution & Development* **5**:579–592 DOI 10.1046/j.1525-142x.2003.03063.x.

**Ortmann AE. 1921.** *A monograph of the naiades of Pennsylvania*. Washington: Board of Trustees of the Carnegie Institute.

**O'Hanlon JC, Holwell GI, Herberstein ME. 2014.** Pollinator deception in the orchid mantis. *The American Naturalist* **183**:126–132 DOI 10.1086/673858.

**Pagel M. 1999.** Inferring the historical patterns of biological evolution. *Nature* **401**:877–884 DOI 10.1038/44766.

**Palmer DH, Kronforst MR. 2020.** A shared genetic basis of mimicry across swallowtail butterflies points to ancestral co-option of doublesex. *Nature Communications* **11(1)**:6 DOI 10.1038/s41467-019-13859-y.

**Parmalee PW, Bogan AE. 1998.** *Freshwater mussels of Tennessee*. USA: University of Tennessee Press.

**Pasteur G. 1982.** A classificatory review of mimicry systems. *Annual Review of Ecology and Systematics* **13**:169–199 DOI 10.1146/annurev.es.13.110182.001125.

**Patterson MA, Mair RA, Eckert NL, Gatenby CM, Brady T, Jones JW, Simmons BR, Devers JL. 2018.** *Freshwater mussel propagation for restoration.* Cambridge University Press.

**Peterson BK, Weber JN, Kay EH, Fisher HS, Hoekstra HE. 2012.** Double digest RADseq: an inexpensive method for de novo SNP discovery and genotyping in model and non-model species. *PLOS ONE* **7(5)**:e37135 DOI 10.1371/journal.pone.0037135.

**Prout T. 2000.** How well does opposing selection maintain variation? *Evolutionary Genetics from Molecules to Morphology.* Cambridge: Cambridge University Press, 157–181.

**R Core Team. 2018.** R: a language and environment for statistical computing. *Available at* https://www.r-project.org.

**Randall JE. 2005.** A review of mimicry in marine fishes. *Zoological Studies-Taipei* **44**:299.

**Robertson MS, Winemiller KO. 2001.** Diet and growth of smallmouth bass in the devils river. *Texas the Southwestern Naturalist* **46(2)**:216–221 DOI 10.2307/3672533.

**Satterthwaite FE. 1946.** An approximate distribution of estimates of variance components. *Biometrics Bulletin* **2(6)**:110 DOI 10.2307/3002019.

**Schaefer HM, Ruxton GD. 2009.** Deception in plants: mimicry or perceptual exploitation? *Trends in Ecology & Evolution* **24(12)**:676–685 DOI 10.1016/j.tree.2009.06.006.

**Sheppard PM, Turner JRG, Brown K, Benson W, Singer M. 1985.** Genetics and the evolution of Muellerian mimicry in Heliconius butterflies. *Philosophical Transactions of the Royal Society of London. B, Biological Sciences* **308(1137)**:433–610 DOI 10.1098/rstb.1985.0066.

**Shine R, Brown GP, Goiran C. 2022.** Frequency-dependent Batesian mimicry maintains colour polymorphism in a sea snake population. *Scientific Reports* **12(1)**:4680 DOI 10.1038/s41598-022-08639-6.

**Smith GR, Taylor JN, Grimshaw TW. 1981.** Ecological survey of fishes in the Raisin River drainage. *Michigan Michigan Academician* **13**:275–305.

**Smits AJ. 2019.** Undulatory and oscillatory swimming. *Journal of Fluid Mechanics* **874**:P1 DOI 10.1017/jfm.2019.284.

**Soltis DE, Morris AB, McLachlan JS, Manos PS, Soltis PS. 2006.** Comparative phylogeography of unglaciated eastern North America. *Molecular Ecology* **15**:4261–4293 DOI 10.1111/j.1365-294X.2006.03061.x.

**Stamatakis A. 2014.** RAxML version 8: a tool for phylogenetic analysis and post-analysis of large phylogenies. *Bioinformatics (Oxford, England)* **30(9)**:1312–1313 DOI 10.1093/bioinformatics/btu033.

**Stein BA, Morse LE. 2000.** A remarkable array: species diversity in the United States. In: *Precious Heritage.* Oxford: Oxford University Press DOI 10.1093/oso/9780195125191.003.0009.

**Surber EW. 1941.** A quantitative study of the food of the smallmouth black bass, *Micropterus dolomieu*, in three Eastern streams. *Transactions of the American Fisheries Society* **70**:311–334 DOI 10.1577/1548-8659(1940)70[311:AQSOTF]2.0.CO;2.

**Thrasher DJ, Butcher BG, Campagna L, Webster MS, Lovette IJ. 2018.** Double-digest RAD sequencing outperforms microsatellite loci at assigning paternity and estimating relatedness: a proof of concept in a highly promiscuous bird. *Molecular Ecology Resources* **18(5)**:953–965 DOI 10.1111/1755-0998.12771.

**Timmermans MJTN, Srivathsan A, Collins S, Meier R, Vogler AP. 2020.** Mimicry diversification in *Papilio dardanus* via a genomic inversion in the regulatory region of *engrailed—invected.* *Proceedings of the Royal Society B: Biological Sciences* **287(1926)**:20200443 DOI 10.1098/rspb.2020.0443.

**U.S. Geological Survey. 2022.** National hydrography dataset (ver. USGS national hydrography dataset best resolution (NHD) for Hydrologic Unit (HU) 4-2001 (published 20191002)). *Available at* https://www.usgs.gov/national-hydrography/access-national-hydrography-products (accessed 17 July 2022).

**VanTassel NM, Morris TJ, Wilson CG, Zanatta DT. 2021.** Genetic diversity maintained in comparison of captive-propagated and wild populations of *Lampsilis fasciola* and *Ptychobranchus fasciolaris* (Bivalvia: Unionidae). *Canadian Journal of Fisheries and Aquatic Sciences* **78(9)**:1312–1320 DOI 10.1139/cjfas-2020-0373.

**Wacker S, Larsen BM, Jakobsen P, Karlsson S. 2018.** High levels of multiple paternity in a spermcast mating freshwater mussel. *Ecology and Evolution* **8**:8126–8134 DOI 10.1002/ece3.4201.

**Warren ML, Burr BM. 1994.** Status of freshwater fishes of the united states: overview of an imperiled fauna. *Fisheries* **19**:6–18.

**Welsh JH. 1933.** Photic stimulation and rhythmical contractions of the mantle flaps of a lamellibranch. *Proceedings of the National Academy of Sciences of the United States of America* **19**:755–757 DOI 10.1073/pnas.19.7.755.

**Winn HE. 1958.** Comparative reproductive behavior and ecology of fourteen species of darters (Pisces-Percidae). *Ecological Monographs* **28**:155–191 DOI 10.2307/1942207.

**Witt WC, Wen L, Lauder GV. 2015.** Hydrodynamics of C-start escape responses of fish as studied with simple physical models. *Integrative and Comparative Biology* **55(4)**:728–739 DOI 10.1093/icb/icv016.

**Zale AV, Neves RJ. 1982.** Fish hosts of four species of lampsiline mussels (Mollusca: Unionidae) in Big Moccasin Creek. *Virginia Canadian Journal of Zoology* **60**:2535–2542 DOI 10.1139/z82-325.

**Zanatta DT, Fraley SJ, Murphy RW. 2007.** Population structure and mantle display polymorphisms in the wavy-rayed lampmussel, *Lampsilis fasciola* (Bivalvia: Unionidae). *Canadian Journal of Zoology* **85(11)**:1169–1181 DOI 10.1139/Z07-089.

**Zanatta DT, Harris AT. 2013.** Phylogeography and genetic variability of the freshwater mussels (Bivalvia: Unionidae) Ellipse, *Venustaconcha ellipsiformis* (Conrad 1836), and Bleeding Tooth, *V. pleasii* (Marsh 1891). *American Malacological Bulletin* **31**:267–279 DOI 10.4003/006.031.0206.

**Zanatta DT, Murphy RW. 2006.** Evolution of active host-attraction strategies in the freshwater mussel tribe Lampsilini (Bivalvia: Unionidae). *Molecular Phylogenetics and Evolution* **41(1)**:195–208 DOI 10.1016/j.ympev.2006.05.030.

**Zeyl JN, Malavasi S, Holt DE, Noel P, Lugli M, Johnston CE. 2016.** Convergent aspects of acoustic communication in darters, sculpins, and gobies. In: Sisneros JA, ed. *Fish Hearing and Bioacoustics: An Anthology in Honor of Arthur N.* Cham: Springer International Publishing, 93–120 DOI 10.1007/978-3-319-21059-9_6.