# Peer review of "Polymorphism in the aggressive mimicry lure of the parasitic freshwater mussel Lampsilis fasciola"

_PeerJ, doi:10.7717/peerj.17359_

## Round 0.1 · original submission · Minor Revisions

Overview
This manuscript examines several different questions about the lures used by one species of freshwater mussel (two species for some questions) to attract the host fishes for the parasitic stage of their offspring. A genomic analysis showed that the two main lure types (darter fish and leech) occur within a single species rather than representing different cryptic species. This conclusion was supported by a mixture of lure types in the offspring of a single female. The proportion of lure types in one population in 2017 was similar to samples taken at the same location between 1955 and 1962, suggesting stability in the relative abundance of lure types. A visual analysis of the lures suggested similarity to four sympatric species of darter and to the medicinal leech. An analysis of the patterns of movement of the two types of lures found small quantitative differences, but the authors concluded that that there was not a behavioral component to the polymorphism.

The diversity of methods presented challenges in finding appropriate reviewers. In this case, we were fortunate to find two reviewers with expertise in genome analysis and evolution. One reviewer also has experience in the analysis of polymorphisms and in fish behaviour. I lack expertise in genomic analysis but have considerable experience with animal behavior and ecology of a variety of organisms, including fishes.

Reviewer 2 considered the genetic analyses well supported. Suggestions were primarily related to placing the study in a broader context.

Reviewer 1 had more concerns. This reviewer indicated a need to be much clearer on how the genomic analysis contributed to understanding of the polymorphism. Is it simply confirming that the lure types occur in conspecific individuals or is there something more than this? They raise some issues where the literature review is imprecise. In addition, this reviewer would like to see a more scholarly approach to the literature on polymorphisms, giving credit to the important early researchers who developed the concepts as well as to recent reviews that bring the concept up to date. They also suggest that valid conclusions from the behavioral analyses may be more limited than expressed in the manuscript.

I found the manuscript very interesting but lacking detail in some important areas. Given my background, it is perhaps not surprising that I had more to say about the behavioral section. There were only a few paragraphs where I agreed with Reviewer 1 that the manuscript could be shortened. There were more sections where additional material is needed. You may treat my comments below as if they are a third review, either making appropriate changes or providing a specific explanation if you do not feel that changes are warranted. I have provided a pdf using highlights to indicate problems and inserted comments to explain or to suggest alternative wording. I caught quite a few spelling errors/typos, but almost certainly missed others. Please do a careful spell check on the Word document before resubmitting. Many of my changes are related to a more concise presentation that reduces redundancy. You do not need to include all my comments from the pdf in your rebuttal unless you disagree and are not making the changes suggested.

Although the changes are very numerous, the majority relate to the presentation and not to fundamental changes in analysis or interpretation. I therefore considered them ‘minor revisions’.

Editor’s Comments
Major Concerns
1. The behavioral aspect of the study lacks some important information and a thorough discussion.
Introduction
In the description of the lures, you should refer explicitly to the behavior so that readers understand that movement of the lure is involved and have an idea of what you mean by behavior. You should refer to any prior studies or descriptions of lure behavior and identify research gaps in this area. On L526, you indicate that the behavior of lures has been studied for decades. That research should be summarized in the Introduction and gaps identified. The description should be sufficient for the reader to understand precisely what aspect your measures quantify. Even when I had reached the Discussion of the swimming motion of the models, I was unclear about how your left and right measurements related to swimming pattern and undulation speed. To understand the behavioral possibilities, I wanted to know how the two sides of the mantle relate to the lure and what proportion of the exposed mantle is involved? Is there movement of the whole lure or only undulation? The background should be sufficient that when you refer to synchronization of movements in the Methods, the reader will understand the implications. I suggest that you improve the description and then try it on some behavioral biologists who have not seen the movement, query them to see how well they understand, and then show them some of your videos to see how well they feel the description conveyed the behavior.
Methods
• There are many potential characteristics of behavior that can be measured, though perhaps fewer in mussel lures than in a free-living organism. Your analysis focused only on the timing of movements rather than any other characteristic such as qualitative descriptions of movement (classically referred to as ethograms) or the magnitude of movements. You need to give the reader an idea as to why timing was the focus of the analysis.
• You started the analysis 42 s after the start of the recording (L286). Do you have any evidence that this was sufficient to avoid effects of disturbance due to setting up the apparatus? This time is very short compared to most behavioral studies. One way to check for this would be to see whether the patterns changed over the subsequent 2.8 min that were analysed.
• The mention of undulation suggests that movements were not simultaneous along the mantle. Did you take measurements at a standard position along the mantle? If so, why there?
• How did you define the start and end of a movement? I presume that there would be acceleration and deceleration, so the criteria for stop and start are needed for someone to be able to repeat the study.
• I was not familiar with gait analysis except in the context of human and other vertebrate terrestrial locomotion. When I did a quick check on Google, that is also what came up. I think you need to briefly define gait analysis and provide a reference for it being applied in this context.
• Be more clear and consistent about the derived measures. ‘Intervals between undulations’ may be the time between the end of one movement and the start of the subsequent one, but you need to be explicit. Speed of undulation and proportion of movements synchronized are undefined. For all measures, be explicit on the units. I don’t see how your measures could give speed, which seems to imply movement of a wave along the mantle. Perhaps you mean rate of undulations (e.g., movement starts per second)? Proportion synchronized requires a definition of how close movements on the two mantle sides need to be in order to be considered synchronized. The GLMM results use a new term movement frequency, which also undefined.
• I wonder if the use of non-parametric tests was a response to a non-normal distribution of intervals. If so, standard deviation is not an appropriate measure of variation.
• Sample size was unclear. There were 30 fasciola and 4 cardium, but what was the mix of lure types in the fasciola? What about the potential variation of sub-types within the darter category?
• Is a sample size of 4 individuals really sufficient for a robust comparison?
Results
The Results of the behavioral analyses are both wordy and incomplete.
• You do not need separate sentences to tell the reader where to find the results, and you do not need to start sentences with the statistical tests. But you do need to provide the test statistic and sample size as well as the direction and magnitude of effect. A more concise style is to present the pattern followed by the test name, test statistic, sample size, p-value, and figure in parentheses. Once stated, it is not necessary to repeat the name of the test. For example, ‘Inter-movement intervals were much smaller in L. cardium than in both lure types of L. fasciola (Pairwise Wilcoxon test, Z < ?, n = a,b,c, p < 0.001) and smaller in L. fasciola leech than in L. fasciola darter lures (Z = ?, n = a.b., p <0.01, Fig. 10A).’
• The section is incomplete because you do not mention the duration or synchrony results.
• Consider reorganizing the results so that there is more emphasis on the comparison between lure times in L. fasciola which was your primary objective. For example, ‘Intervals between movements in L. cardium were shorter (stats), less variable (stats), but longer in duration (stats) and much synchronized (stats) than in L. fasciola (Fig. 9). Differences between the lure types of L. fasciola were smaller, with inter-movement intervals in the darter phenotype that were longer (stats) and marginally non-significantly more variable (stats) but similar in duration (stats) and degree of synchronization (stats, Fig. 9).’
• Because a non-significant comparison can be a consequence of sample size or variability, you should not consider that a p-value of 0.09 indicates that they are the same. Of course, you cannot conclude that they are different, but you can highlight the uncertainty.
• Would it be worth starting with a broad, qualitative description of the behavior seen in the videos?
Discussion
• You have rather complex results that require a more nuanced discussion.
The Kruskal-Wallace followed by Wilcoxon revealed differences in some traits but not others. However, the GLM seems to have been completely non-significant, if I understand correctly. Which should you accept and why? What are the implications of this difference? What are the implications of some measures differing while others do not? While there are limits to your ability to confirm differences based on your results, the non-significant trends mean that you also don’t have strong evidence that there is no difference (contrary to a statement in the Abstract). How much of a difference and what kind would you have needed to conclude that there was also a behavioral polymorphism? What are the implications of a quantitative, but potentially real, minor difference between morphs? How much of any potential difference between lure types was captured by your measures?

2. The investigation of potential models also needs some additional information
• The Introduction does not document what is known and unknown about the models for the lures of the study species. It is not enough to merely mention lack of specific information in the list of objectives. Since you developed a specific procedure to try to determine the models, it would be good to review how any other studies have confirmed the models for other mussel species. If there have been no detailed studies, that should be mentioned and you should consider what type of evidence has been used to confirm models for other mimicry systems.
• Having introduced the topic, the objective to determine the model(s) can be more concisely stated.
• Methods: how were the 27 photographed specimens selected? Were they all that you could find? Did you try to maximize the range of visual appearances or simply select haphazardly? Did you include both darter and worm forms? Did you search in all habitats where the mussel is found? These aspects have implications for the interpretation of the numbers of different darter types.
• There is a discrepancy between the Methods and Results. The Methods refers only to 27 photos from one river system, but the first paragraph of Results introduces comparisons from other parts of the species range. It is not clear if this is original work belonging in Results accompanied by a more complete Methods section or if it is from the literature and more appropriate to the Discussion.
• Do you have any suggestions as to whether the variation within the darter is a polymorphism or a continuous variation?
• I was initially surprised that there was such precise mimicry in this system, give the wide range of fishing lures used by recreational fishers for bass. Then, I realized that if infestation by glochidia is an aversive experience, bass might try to avoid mussels, setting up a coevolutionary race between mussels and hosts. The descriptions of how fish respond to infestation in Haag and Warren suggest that this may well be the case. This idea is presumably already in the literature, and you might consider whether it adds anything to the discussion of the precision of the similarity between suggested models and the lures.

Other suggestions
Title: the title is rather long and unclear, attempting to include too many aspects. It is also incorrect; the lures do differ in behavior, albeit not very much. Consider something simpler such as ‘Polymorphism in the aggressive mimicry lure of the parasitic freshwater mussel Lampsilis fasciola’.
Abstract
L30-34. There is substantial redundancy in stating the questions and then the findings. I think that it would be more concise to incorporate the goal into each finding.
L36. Like Reviewer 1, I did not know what a ‘true polymorphism’ referred to. What other kind is there? Might some readers take this be a reference to a variable monomorphism? If what you mean is that it is intraspecific rather than representing two cryptic species, say so. (Also applies to L470.)
L45-46. I would be more comfortable if you indicated that there were some differences between the morphs (as your statistical analysis showed), but that they appeared to be minor. As Reviewer 1 indicated, you found no clear evidence for a behavioral component rather than eliminating the possibility of any difference, given the limited behavioral measures taken.
Introduction
L80. First describe the patterns. I have suggested a more concise presentation. Then, insert a separate sentence to show how this is classified as an aggressive mimic. (Many readers may not be familiar with the term, and it was questioned by Reviewer 1.)
L80. Instructions to authors say to use Fig. for figures except at start of a sentence and to use capital letters for figure panels.
L81. Elaborate on ‘cryptic radiation’ as this is the justification for part of your investigation.
L81-88 are out of place in the Introduction. You are partially introducing results here. I suggest that you mention the presence of lures in males and juveniles, credit authors who noted them, and indicate what was still unknown before your study. Is it only photographic documentation or description as well? Add it to your objectives and devote a subsection of Method, Results and Discussion to the topic. Alternatively, remove these lines and just add the illustrations of rudimentary lures to the Results, along with a note indicating lack of previous published photos, where you describe the female lures.
L121. Should you clarify what ‘molecular phylogenetic corroboration’ means? Are you contrasting to microsatellites or to the limited sample?
L122. The question of persistence comes out of the blue. There should have been some indication of why the question is important and where it has been studied. Also, you are not really investigating the persistence of lures (you provided no evidence of females that lacked lures) but of the persistence of their relative abundance.
L124. Similarly (as noted previously), the topic of models and what evidence there is in this and other species and how generally models are identified belongs in the previous introduction. Revise the statement here to make the objective clear; you now simply assert that little is known and not what you intend to do.
L124. As noted previously, lure behavior needs to be introduced above. Rephrase as a clear question/objective.
L130-146. Delete this because it is redundant to Results. Move the first sentence starting on L129 to the end of the previous paragraph.
L245. It is not clear to me why you use ratios instead of percentages. With different sample sizes, the similarity of ratios is not immediately obvious whereas percentages are clear. This was satisfactorily presented in the Results.
L345ff. I have made a number of suggestions for a much more concise, less redundant presentation of the relative numbers of the two lure types.
L388. I don’t think you need to write out Lampsilis each time you refer to the study species. Once you have identified the genus L. should be sufficient when it is clear that you are talking about your own specimens. (Check throughout the manuscript to be sure that you have a consistent use of this approach.)
L454. Generally, it is not necessary to cite figures in the Discussion if they have already been cited in the Results.
L459ff. This section repeats Methods but is probably adequate here. However, you should also indicate the evidence for geographical structuring as you did for lure types.
L481. Should you indicate what kind of analyses or would this be obvious to more readers readers with a stronger background in genetics?
L490. This reference implies that Jamie would have shown that the mussel system was a case of aggressive mimicry. If you are using it just to provide a definition of the concept, you should go back to the original reference.
L505. It seems odd that in Results you give percentages for the rare morph and in Discussion for the common one. Is there a good reason for this?
L517. It seems relevant to include the total number of sampling locations here.
L531. If this is your own observation, it should be included in Results.
L533. Specify what you think are the similarities and also the differences. Presumably, the continuous undulation is a difference.
L537. How did you conclude that there was no polymorphism in cardium? With a sample size of four, isn’t this a bit premature? Perhaps you need to cite the literature here.
L542. You might consider either your own observations of darters in the field if you saw some when photographing mussels or look at some of the better YouTube videos such as https://www.youtube.com/watch?v=zYkk2Tw31OU to better describe how darters move in current and still water. There may also be some literature on their movements. In my superficial look at the videos, they seem to sometimes keep their body more or less still or undulating slightly in the current but jump ahead using pelvic fins and sometimes body-caudal fin propulsion.
L550, 555. The central pattern generator was unrelated to your study and entirely speculative here; I think it should be left out.
L552. Is it possible that leeches move differently when moving along the substrate rather than swimming in midwater?
L554. Note that the real differences between the morphs also need explanation. Could they be part of the genetic control system, even though minor?
References: Please check all references carefully. I noted some lack of italics, capital letters in article titles, and incomplete references. Do not assume that I highlighted all occurrences.
Figures
I have made a number of suggestions for more concise captions. Note PeerJ requirement for capital letters for panels.
Fig. 5. The large gap between 1962 and 2018 is not very informative. I suggest breaking the time axis with // and using wider bars. Fonts on both axes are likely too small. Using a photocopier, reduce the figure to the size it is likely to appear on a pdf and see if the numbers and words are still readable.
Fig. 7. Nice illustration. (In fact, all the illustrations are nicely done, but I was particularly struck by this one.) The label fonts may be too small, but I will leave that to PeerJ to decide.
Fig. 8. Add to the caption at the end something like: ‘Note the different spatial scales of the panels.’
Fig. 9. Consider adding a real time scale to the x-axis in addition to frame number. Fonts are likely to be too small.
Fig. 10. Boxplots can vary in what they show. Be explicit about what the middle line, gray rectangle, whiskers, and dots show. For panels A and B, outliers force excessive compression of the main data. I suggest that you rescale to maximum mean of 5.0 and SD of 5.0 and include in the caption the values of outliers not shown. For panel C, include zero on the y-axis to make it clearer. For panel D, the line for cardium at or near proportion 1.0 is not easily noticed. Leave more white space above 1.0 and add 0.0 to y-axis. To avoid small fonts, reduce the words on the y-axis labels and include full wording in the caption. For example, A. Mean interval (s), B. S.D. of interval (s), C. Mean duration (s), D. Proportion synchronized. Note backwards quote marks on leech.
Table 1. This table looks more appropriate for supplementary material, but I am not a good judge of this because of my limited experience with genomics.
Table 2 mostly duplicates a figure. Move to Supplementary material.
Supplementary figures
It is not clear why the data sheet (Supp. Fig. 3) is needed since the relevant information was reported in the manuscript.

Reviewer 1 ·

Basic reporting

Well-written
Unsatisfactory familiarity with the literature on polymorphisms
Article structure good

Experimental design

no comment

Validity of the findings

no comment

Additional comments

General;
- this is an interesting ms and warrants publication but appears to be much too long for the amount of new content provided. It is basically confirmatory to Zanatta’s 2007 ms using newer methods. For example, the detailed genomic methodology and results do not clearly contribute to the theme of the title and ms. That different populations along a river differ in genomic structure is expected but what does this have to do with the lure polymorphism. I suspect the genomics can be greatly reduced.

Line 30: ”….. lacks those features and has an orange and black coloration. We investigated this phenomenon to 1) confirm that it is a true polymorphism
…… behavioral component. Detection of within-brood lure variation and within-population phylogenomic (ddRAD-seq) analyses of individuals bearing different lures confirmed that this phenomenon is a true polymorphism…”

Response: I don’t understand what is meant by a ‘true polymorphism’. The extensive literature on these mussels clearly indicates that it is a ‘polymorphism’, a term originally defined in 1940 by EB Ford. A polymorphism can be either genetic or environmentally determined. Perhaps the authors meant that they wished to determine whether it is ‘genetic polymorphism? But without any analyses of cohorts with known parentage, this is not readily determined. The genomic data that are present gave little insight on this issue..

Line 62: “…… Mimetic systems that are polymorphic (multiple within-species mimic morphs with discrete models) have been particularly influential in uncovering the genetic basis of complex adaptive traits in natural populations (Jay et al., 2018; Palmer & Kronforst, 2020). Such polymorphisms are rare in nature, with the most well studied examples occurring in papilionid butterflies (Hazel, 1990; Joron & Mallet, 1998; Nijhout, 2003; Jay et al., 2018; Palmer & 67 Kronforst, 2020).”

Response: Jay et al and Palmer etc are not appropriate references for the sentence. Please consult the early work of Sheppard and Clarke in the 1960’s of Papilio, also extensively discussed in Ford’s book on Ecological Genetics.
77: “…. 1933): a pigmented fleshy extension that acts as an aggressive mimic (Jamie, 2017)”
Response: I recognize that this is a terminology bused by Jamie but it is not clearly stated why the lure polymorphism would be considered an ‘aggressive mimic’ rather than ‘mimic’.It implies something of the behaviour of the mimic which is not the case here. This should be better explained.
145 “….non-polymorphic fish-mimic lures. We conclude that the L. fasciola mantle lure polymorphism does not include a significant behavioral component.”

Response: I suspect that it would be safer to say that ‘no behavioral differences were detected’ as there were only several behavioural proxies quantified.

198: “…..Phylogenomic analyses.. ‘:
Response: I recognize that inclusion of genomic data to most current mss are useful but it has not been clearly explained how the genomic data contributes to the interpretation of the lure polymorphism.

286 “….frames to avoid any initial setup effects on mussel display behavior. The frame numbers when an individual movement began and ended were noted, and movements of the left and the right mantle lure flaps were assessed separately”

Response: Did this involve any interaction with the smallmouth bass for any of the video sequences?
483: “….In butterflies, polymorphic mimicry in wing pigmentation is controlled by an
introgressed mimicry supergene in Heliconius species (Jay et al., 2018) and by mimicry alleles of the transcription factor doublesex (dsx) in some Papilio species (Palmer & Kronforst, 2020)”.
.
Response: There is a well-published history of the genetics of Heliconius polymorphisms that precede by 5 decades these references. Jay et al are confirmatory for what was done before. Use original when possible.
486 “….Note, however, that the Lampsilis fasciola mantle lure mimicry polymorphism differs in important ways from these butterfly systems. It is more complex because it involves putative models (darters and leeches) from disparate phyla rather than from similar morphospecies (other butterflies), thereby requiring polymorphic trait differentiation in pigmentation and in morphology (Figure 1b)”

Response: Not correct as the Batesian and Mullerian butterfly mimics have multiple models and can differ in flight patterns, timing, microhabitat use etc. See Ford 1963.
496:”…. One hypothesized mechanism for the persistence of
polymorphisms in a species or population is frequency-dependent selection, where selection for rare phenotypes are selected cause the ratio of phenotypes to vary over time (Ayala & Campbell,498 1974) and lines 501 and 502)”.

Response: This is not correct as stated, . Frequency dependence does not refer to varying frequencies over time although it can lead to this. Rather it is simply the inverse relationship between frequency and fitness. Sex ratio is such an example. As well, Bryan Clarke (1964) was one of the first to introduce the role of frequency dependence with respect to polymorphic traits and should be referenced here rather than a recent article…perhaps cite this as a review
.
504: “…However, the historical (1954-1962) and contemporary (2017) data from Sharon
Mills County Park (Figure 5) did not show evidence of such oscillation.
Response: Stability in frequencies over time is just as likely to be due to frequency dependence as are fluctuations over time. Sex ratio is an example of stability yet is generally assumed to be maintained by frequency-dependence.

507: “….Theoretically, there are other mechanisms for balancing selection to maintain polymorphisms over long time-scales, including heterozygote advantage or opposing
selection pressures favoring different alleles at polymorphic loci (Prout, 2000; Mérot et al.,(2020)”.

Response: The statement is certainly correct. It is also conceptually broad and requires citations relevant to the concepts. See Ford 1963 for development of these ideas. Prout and Merot have done interesting research but are not responsible for the conceptual development of the ideas.
521: “…..Another possibility is that at least some L. fasciola .darter-like. lures across the mussel.s range are composite mimics of visual elements from more than one member of their local darter fauna, but that remains to be established….”..

Response: This is certainly highly reasonable but I am surprised that the authors did not raise the prospects of other mechanisms such as plasticity, for example mussel detection of chemical cues that differentiate the models. Apart from the within-brood morph frequencies, are there any other data that allow you to exclude phenotypic plasticity?. There are several lepidopteran caterpillar polymorphisms that have such plasticity to complex cues (see DAS Smith).

537 “…… there appears to be no behavioral polymorphism to distinguish the darter from leech lure phenotypes...”

Response: The short duration of the behavioural observations might warrant ‘no detectable behavioral……’ rather than ‘no behavioral polymorphism’.

552 “….different from the lure undulations observed in the leech-like L. fasciola lures. Despite small differences in overall lure beat frequency between .darter. and .leech. mimics, it seems as though the polymorphic mimicry is only skin deep. The rhythmic movements of lure undulations..”

Response: This seem to overextend the results. Was there a chemical analyses of the two lure types that would justify the metaphor of ‘skin-deep’? In other words, can chemical signals be excluded. That fish have such exceptional chemosensory modality could suggest the potential.
Figure 10: These are two visual attributes. Presumably there are multiple additional visual attributes of the behavior.

Reviewer 2 ·

Basic reporting

I really enjoyed this study, and I am impressed by the combination of phylogenomics, behavioral assays, field work, and captive breeding. The results are clear and genetic results follow standard techniques in the field. Analyses seem well supported.

The genetic basis of mussel lures is poorly understood. These analyses clarify the genetic basis of rapidly evolving traits that are important for parasite-host coevolution. The mimicry of other organisms in mantle lures adds another layer of co-evolutionary complexity. As a case study for connecting genotype to phenotype this is an excellent piece of work.

The introduction briefly discusses mimicry in other organisms as a concept, but focuses fairly squarely on butterflies. There are lots of different cases of mimicry in nature across the tree of life (e.g. mantids, Myrmarachne, poison frogs, caterpillars, anglerfish, snapping turtle tongues). I don't have strong opinions on which specific systems, but broadening the discussion to make it more complete would help place this paper properly in the field. This context may need a similar paragraph in the Discussion to clarify the broader context of the results and how they relate to evolution.

Related, some information about mussels is given in discussion, but would be helpful if added to Results to offer context as results are presented. If the authors hope to reach the community working on other organisms, it will be helpful to couple some near scope interpretation with the data.

Experimental design

The experiments are all designed well to clarify associations in mantle lures. The results are clear and the study reads simply. (Though I'm certain it wasn't simple to implement~!) Work in this system is extensive with field collection, captive breeding, and phenotypic + genetic assays.

Validity of the findings

Well designed, well implemented.

Additional comments

I enjoyed this study and will be glad to see it out for the community to read.

---

## Round 0.2 · accepted · Accept

This manuscript and the rebuttal letter show careful and thoughtful responses to the numerous comments provided by reviewers and editor. I have attached a pdf with a number of minor spelling and grammatical corrections that can be added during manuscript processing.